# Self-Evolving Multi-Agent Collaboration Networks for Software Development

**Yue Hu[1], Yuzhu Cai[2,3], Yaxin Du[1], Xinyu Zhu[1], Xiangrui Liu[1], Zijie Yu[1],**
**Yuchen Hou[1], Shuo Tang[1], Siheng Chen[1,3]**
[1] Shanghai Jiao Tong University, [2] Beihang University, [3] Shanghai AI Laboratory
[1] {1867112931,sihengc}@sjtu.edu.cn, [2] caiyuzhu@buaa.edu.cn

## Abstract

LLM-driven multi-agent collaboration (MAC) systems have demonstrated impressive capabilities in automatic software development at the function level. However, their heavy reliance on human design limits their adaptability to the diverse demands of real-world software development. To address this limitation, we introduce EvoMAC, a novel self-evolving paradigm for MAC networks. Inspired by traditional neural network training, EvoMAC obtains text-based environmental feedback by verifying the MAC network's output against a target proxy and leverages a novel textual backpropagation to update the network. To extend coding capabilities beyond function-level tasks to more challenging software-level development, we further propose rSDE-Bench, a requirement-oriented software development benchmark, which features complex and diverse software requirements along with automatic evaluation of requirement correctness. Our experiments show that: i) The automatic requirement-aware evaluation in rSDE-Bench closely aligns with human evaluations, validating its reliability as a software-level coding benchmark. ii) EvoMAC outperforms previous SOTA methods on both the software-level rSDE-Bench and the function-level HumanEval benchmarks, reflecting its superior coding capabilities. The benchmark can be downloaded at https://yuzhu-cai.github.io/rSDE-Bench/.

## 1 Introduction

Automatic software development focuses on generating code from natural language requirements. Code is a universal problem-solving tool, and this automation presents significant potential to provide substantial benefits across all areas of our lives Li et al. (2022a). Recently, the industry has introduced several large language model (LLM)-driven coding assistants, including Microsoft's Copilot Microsoft (2023), Amazon's CodeWhisperer Amazon (2022), and Google's Codey Google (2023). These coding assistants significantly advance human efficiency and yield considerable commercial benefits. Despite the initial success of LLMs in assisting with line-level coding, they struggle to tackle more complex coding tasks. This limitation stems from the restricted reasoning abilities of single LLMs and their lack of capacity for long-context understanding Wang et al. (2024a); Li et al. (2024a); Wang et al. (2024b).

To handle function-level coding tasks, numerous multiple language agent collaboration (MAC) systems have been proposed Li et al. (2023); Hong et al. (2023); Chan et al. (2024); Islam et al. (2024); Yang et al. (2024b); Li et al. (2022b); Osika (2023). These MAC systems function as LLM-driven agentic workflow. They follow human-designed standardized operating procedures to divide the complex coding tasks into simpler subtasks within the workflow, allowing each agent to conquer specific subtasks. These MAC systems significantly advance coding capabilities from line-level to function-level tasks. However, current MAC systems rely on heuristic designs. These human-crafted static systems have two inherent limitations: i) their performance is confined to human initialization. Given the diversity of real-world coding tasks, human design cannot fully address the specific needs of each task; and ii) they lack the flexibility to adapt to new tasks. This rigidity necessitates that researchers and developers manually decompose tasks and create prompts. The complexity of this process inhibits effective human optimization for adapting to new challenges.

To address these limitations, we present EvoMAC, a novel self-evolving paradigm for MAC networks. EvoMAC's key feature is its ability to iteratively adapt both agents and their connections during test time for each task. Inspired from the standard neural network training, the core idea of self-evolution is to obtain text-based environmental feedback by verifying the MAC network's generation against a target proxy, then leverage a novel textual back-propagation to update the MAC network. Following this general paradigm, we specify EvoMAC for software development, which comprises three essential components: i) an adaptable MAC network-based coding team that generates code through feed-forward; ii) a specifically designed testing team that creates unit test cases serving as the target proxy and verifies the generated code in the compiler to produce objective feedback; and iii) an updating team that uses the textual back-propagation algorithm to update the coding team. By cycling these three components, the coding team can iteratively evolve and generate codes that are better aligned with the unit test cases, eventually fulfilling more requirements of the coding task.

Our self-evolving MAC network has the potential to further advance coding capabilities from function-level to more complex software-level tasks. As it can iteratively address lengthier task requirements and cater to realistic software development demands. However, existing benchmarks typically focus on specific individual functions Chen et al. (2021); Austin et al. (2021); Yang et al. (2024a); Khan et al. (2023) or bug-fixing Jimenez et al. (2023), leaving a significant gap in providing comprehensive requirements for software development. This gap makes it difficult to fully assess the potential of our self-evolving MAC network.

To support the development of software-level coding capabilities, we propose rSDE-Bench, a novel requirement-oriented software development benchmark. It is the first benchmark that features both complex and diverse software requirements, as well as the automatic evaluation of requirement correctness. rSDE-Bench involves 53 coding tasks with 616 requirements, covering two typical software types, Website, and Game, and two requirement difficulty levels, Basic and Advanced. Each coding task consists of two components: i) multiple requirements that clearly outline measurable software functionalities, item by item, and ii) paired black-box test cases that automatically verify the correctness of each requirement. rSDE-Bench can achieve automatic evaluation with these synchronized pairs of requirements and test cases. The rSDE-Bench introduces new software-level challenges, including lengthy requirement analysis and long-context coding, which are essential in real-world software development but are absent in existing benchmarks.

To validate the effectiveness of our proposed EvoMAC and rSDE-Bench, we conduct three key evaluations. First, we compare our automatic evaluation in rSDE-Bench with human evaluation, achieving a coherence score of 99.22%, demonstrating its reliability. Second, we compare EvoMAC against five multi-agent and three single-agent baselines. EvoMAC significantly outperforms previous SOTAs by 26.48%, 34.78%, and 6.10% on Website Basic, Game Basic, and HumanEval, respectively, underscoring its effectiveness. Third, we evaluate EvoMAC with varying evolving times and two different driving LLMs. The results indicate that EvoMAC consistently improves with more evolving times and shows convincing enhancements regardless of the driving LLM used, further demonstrating the effectiveness of our self-evolving design.

To sum up, our contributions are:

• We propose EvoMAC, a novel self-evolving MAC network, and apply it to software development. EvoMAC can iteratively adapt both agents and their connections during test time for each task.

• We propose rSDE-Bench, a novel requirement-oriented software development benchmark. It is the first benchmark that features both complex and diverse software requirements, as well as the automatic evaluation of requirement correctness.

• We conduct comprehensive experiments and validate that: automatic evaluation in rSDE-Bench is highly aligned with human evaluation; EvoMAC outperforms previous SOTAs, and self-evolving promises continuous improvement with evolving times.

## 2 RELATED WORKS

**LLM-based multi-agent collaboration.** LLM-driven multi-agent collaboration (MAC) systems Xu et al. (2023); Hua et al. (2023); Ziems et al. (2024); Wu et al. (2023); Hong et al. (2023); Chan et al. (2024); Mandi et al. (2024a) enable multiple agents to share information and collaboratively complete the overall task. These MAC systems function as agentic workflows. They have demonstrated enhanced problem-solving capabilities in various domains, such as mathematics Islam et al. (2024),

software development Qian et al. (2023); Hong et al. (2023), embodied task Mandi et al. (2024b) and social simulation Ziems et al. (2024); Pang et al. (2024); Li et al. (2024b). However, these systems Wu et al. (2023); Chen et al. (2023) heavily rely on manually designed workflows, which lack generalizability and the labor-intensive nature of manual design poses significant limitations. To address this issue, we propose a novel self-evolving paradigm, which allows agents to update and improve through external feedback, enabling dynamic adaptation and more advanced performance across varied tasks.

**Software development benchmarks.** Software development benchmarks aim to evaluate models in the task of generating code from natural language descriptions Zheng et al. (2023). These benchmarks typically include task definitions and evaluation criteria. Existing benchmarks can be categorized into three types: i) function completion (HumanEval Chen et al. (2021), MBPP Austin et al. (2021), EvalPlus Liu et al. (2023), xCodeEval Khan et al. (2023)); ii) bug repair (SWE-bench Jimenez et al. (2023)); and iii) software generation (SRDD Qian et al. (2023), SoftwareDev Hong et al. (2023)). Function completion and bug repair benchmarks are confined to function-level task definitions, missing the diverse realistic software requirements. Software generation benchmarks often depend on expensive human evaluations or indirect similarity-based measurements, unable to automatically and accurately verify the requirement correctness. To address these limitations, we introduce rSDE-Bench, the first benchmark contains both diverse software requirements and automatic evaluation of requirement correctness. It can support the development of more realistic software-level coding capabilities.

## 3   EVOMAC: SELF-EVOLVING MULTI-AGENT COLLABORATION NETWORK

This section presents EvoMAC, a novel self-evolving multi-agent collaboration network and its application to software development. The key feature of EvoMAC is its ability to iteratively adapt both agents and their connections during test-time for each task, mimicking the back-propagation process, a core algorithm in neural network training. We first formulate a general self-evolving paradigm in Sec. 3.1 and then describe its application to software development in Sec. 3.2.

### 3.1   A GENERAL SELF-EVOLVING PARADIGM VIA TEXTUAL BACKPROPAGATION

**Multi-agent collaboration network.** A multi-agent collaboration (MAC) network is a computational graph representing agentic workflows, where multiple agents empowered by LLMs interact as interconnected nodes to coordinate and share information for complex task-solving. The intuition behind to divide the complex task into more specific and manageable subtasks for each agent, allowing the overall task to be gradually conquered through the agentic workflow. Mathematically, we represent a MAC network with $N$ autonomous agents as a directed acyclic graph $\mathcal{A} = (\mathcal{V}, \mathcal{E})$, where $\mathcal{V} = \{v_i\}_{i=1}^{N}$ is the set of $N$ nodes, and $\mathcal{E} = \{e_{i,j}\}_{i,j \in [1,...,N], i \neq j}$ is the set of directed edges with no circles. The $i$-th node $v_i$ represents the $i$-th autonomous agent with the prompt $p_i$, which specifies its subtask. The edge $e_{i,j}$ represents the task dependency between the $i$-th agent and the $j$-th agent, indicating that the $j$-th agent's subtask should be executed after the $i$-th agent's subtask in the agentic workflow. The overall graph topology specifies the agentic workflow. Analogy to traditional neural networks, agents function similarly to neurons, with agent prompts serving as neurons' weights and the agentic workflow as the layers and connections.

The feed-forward pass of MAC network is the execution of the agentic workflow. In this process, each agent is given two inputs: the initial task requirement and the output from the previous agent. Using these, each agent produces an output that fulfills its specific subtask. Eventually, the last agent's generation constitutes the final output, integrating all completed subtasks. Note that the initial task requirement is input to each agent as context, providing supplementary details to aid in the implementation of each subtask.

Recently, various MAC networks have been designed using human expertise to assign fixed agent prompts and workflows Hong et al. (2023); Chan et al. (2024), resembling untrained neural networks. However, these designs solely rely on human priors and lack adaptability, causing limited performance improvement over a single agent. To overcome this, inspired by neural network training, we propose a self-evolving paradigm for multi-agent collaboration networks, enabling both agents and their connections to dynamically evolve during test-time for each given task.

**Optimization problem.** Here we consider a general generation task. During test-time, given a task, the MAC network performs a feed-forward pass to generate the final output without knowing its

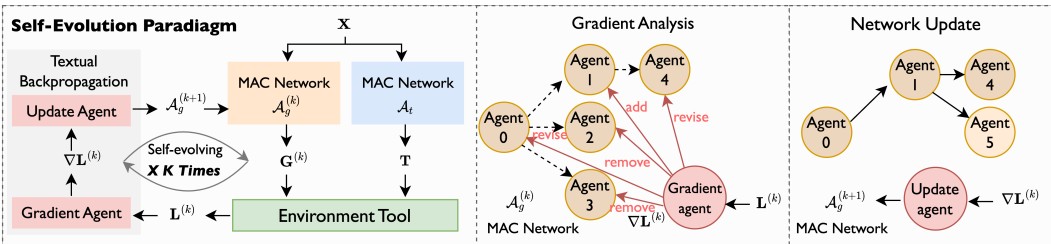

Figure 1: The general self-evolving paradigm.

quality. The key to evolution during test-time is to set up a target proxy for the MAC network to guide its improvements in the generated output. Here we consider this target proxy as the conditions for task completion, such as unit tests in coding, and we can produce such a target proxy by another group of autonomous agents based on the same task description. Then, the quality of each generated output can be verified according to the target proxy. This approach relies on two key assumptions: (i) generating a target proxy is significantly simpler than completing the original generation task, and (ii) the generated output can be correctly verified against the target proxy through an objective environment. These assumptions are practical in many applications. For example, in code generation, producing unit tests, the expected input-output pairs, is much easier than generating the entire code; meanwhile, a code compiler naturally acts as the objective environment to check the correctness of the generated code against the unit test, providing objective and informative feedback.

Mathematically, let $\mathbf{X}$ be the textual description of a task. Given the MAC network $\mathcal{A}_g$, the generated output is $\mathbf{G} = \Phi(\mathbf{X}, \mathcal{A}_g)$, where $\Phi(\cdot, \cdot)$ is the general feed-forward operator that executes the agentic workflow, processing the input text through the MAC network. Similarly, the target proxy is $\mathbf{T} = \Phi(\mathbf{X}, \mathcal{A}_t)$, where $\mathcal{A}_t$ is another MAC network designed for producing the target proxy. Note that we aim to evolve and optimize $\mathcal{A}_g$, while keep $\mathcal{A}_t$ predefined and fixed. The optimization of our self-evolution is formulated as,

$$\mathcal{A}_g^* = \min_{\mathcal{A}_g} \langle \Phi(\mathbf{X}, \mathcal{A}_g), \mathbf{T} \rangle_E, \ \ \text{subject to:} \ \ \mathbf{T} = \Phi(\mathbf{X}, \mathcal{A}_t), \tag{1}$$

where $\langle \cdot, \cdot \rangle_E$ is an objective environment executor that receives the generated output and the target proxy as inputs and outputs a text-based environmental feedback. Akin to the loss function in traditional neural network training, which quantifies the difference between the generated output and the ground-truth, the objective in (1) evaluates whether the generated output meets the conditions of the task completion using the environment, subsequently producing execution reports as the text-based environmental feedback. Here the minimization operation $\min$ is defined to reduce the failures during execution. With the guidance of the target proxy and the objective feedback given by the environment, the MAC network can improve its success rate of task completion during test time.

Note that, another straightforward way to enable the MAC network's evolution is through the self-critique strategy Zhou et al. (2024); Valmeekam et al. (2023); Xu et al. (2024); Asai et al. (2023), which employs a critique agent to assess the generated output directly. This approach has two inherent limitations: i) the critique may be subjective and biased, and ii) the critique agent can have hallucinations, causing inconsistencies and errors. These limitations can cause the MAC network to become entrenched in its own preferences or evolve in the wrong direction, especially iterating multiple times; see our experimental validations in Tab. 2. In comparison, our approach leverages an environment executor to provide objective feedback, preventing bias and hallucinations.

While we use the analogy between our self-evolution process and neural network training for motivating, they are significantly different in three key aspects: (i) our self-evolution occurs at test time without a dedicated training phase; (ii) it evolves for each specific task individually rather than over a batch of samples; and (iii) the environmental feedback are usually texts, not be numerical values, which cannot be optimized by the standard backpropagation. This motivates us to propose our textual backpropagation.

**Solution based on textual backpropagation.** The self-evolution solution iteratively updates the MAC network using a textual backpropagation algorithm, guided by the environmental feedback. The core idea is to analyze the influence of each agent in the MAC network $\mathcal{A}_g$ to the final environmental feedback and use these analyses to update the agent prompts and the agentic workflow in $\mathcal{A}_g$. This

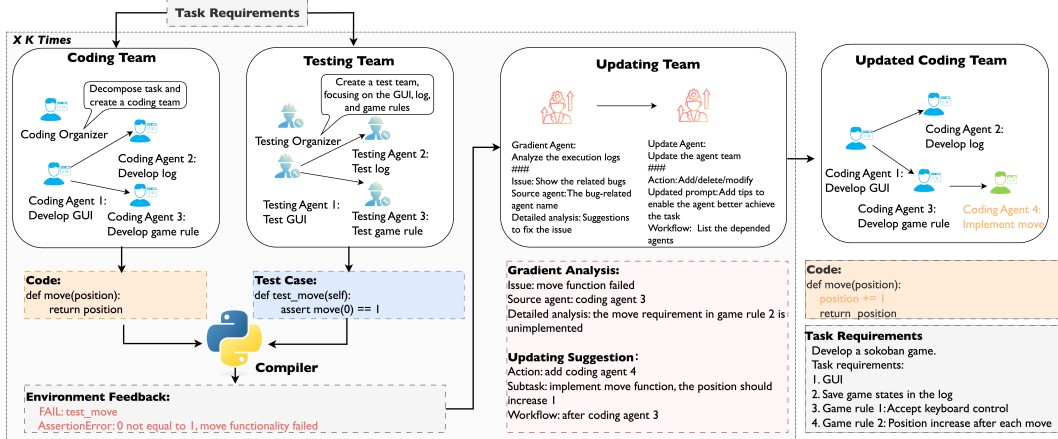

Figure 2: EvoMAC takes task requirements as input and iteratively updates the coding team to generate code that better fulfills the requirements.

is achieved by two collaborative agents, each responsible for one of the two key steps: (i) textual gradient analysis and (ii) network update. The overall algorithm can refer to Alg. 1 in the appendix.

First, the gradient agent takes the environmental feedback as the input and outputs textual gradients that describe the impact of each agent in the MAC network. Let $\mathcal{A}_g^{(k)}$ and $\mathbf{L}^{(k)}$ be the MAC network and the environment feedback at the $k$-th iteration. The textual gradient is then $\nabla \mathbf{L}^{(k)} = \mathcal{G}(\mathcal{A}_g^{(k)}, \mathbf{L}^{(k)})$, where $\mathcal{G}(\cdot, \cdot)$ is the gradient analysis operator managed by the gradient agent; see its prompt in Appendix. The textual gradient details three-fold information for each agent inside $\mathcal{A}_g^{(k)}$: 1) whether this agent's subtask is fulfilled; 2) whether this agent introduces errors; and 3) whether any subtask is missed in the current MAC network.

Second, based on the textual gradients, the updating agent iterates the MAC network as $\mathcal{A}_g^{(k+1)} = \mathcal{U}(\mathcal{A}_g^{(k)}, \nabla \mathbf{L}^{(k)})$, where $\mathcal{U}(\cdot, \cdot)$ is the updating operator managed by the updating agent. This operator guides the updates from three-folds: 1) removing the agents whose subtasks have been completed; 2) revising the erroneous agent's prompts by adding potential solutions provided in the gradient analysis; and 3) adding new agents for missing subtasks and restructuring the workflows based on the subtask dependencies noted in the gradient analysis; see the prompt details in Appendix. These adjustments address existing issues and fulfill unmet requirements in the current generation of the MAC network, promising improvements in the updated version.

Note that, the key of the textual backpropagation is the prompt designs for both gradient analysis and network updates. The design must i) thoroughly evaluate the subtask of each agent in the MAC network according to the objective environment feedback and determine necessary adjustments to the MAC network to address existing issues, fulfilling the unmet requirements; and ii) maintain coherence, ensuring that issues identified by the gradient agent can be effectively resolved by the updating agent's modifications to the MAC network.

## 3.2 SELF-EVOLUTION FOR SOFTWARE DEVELOPMENT

In this section, we apply the self-evolving paradigm to the task of software development. The overall architecture of the proposed self-evolving multi-agent collaboration network for software development is illustrated in Fig. 1. Given a coding task, the coding team, corresponding to the MAC network $\mathcal{A}_g$, generates all the codes through its forward-pass; the testing team, associated with the MAC network $\mathcal{A}_t$, is responsible for creating the target proxy; that is, unit tests of the coding task; and the objective environment tool is realized through the compiler. The identified bugs during execution form the textual environmental feedback. The updating team, consisting of two collaborative agents, manages the textual backpropagation. By continuously cycling through feed-forward, feedback collection, and textual backpropagation processes, the coding team is iteratively refined to align more closely with the test cases.

Since unit test generation is much easier than the original logical code generation, the testing team usually can produce high-quality test cases, which are closely aligned with the task requirements.

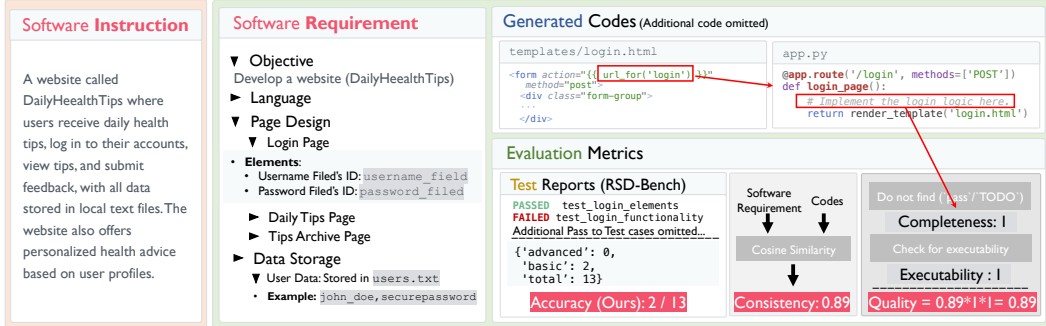

Figure 3: Comparison between instruction-oriented and requirement-oriented evaluations. rSDE-Bench accurately reflects requirement fulfillment with the proposed accuracy score of $2/13$, while the indirection evaluation misjudges with high scores $(0.89)$, failing to detect missing functionality.

Then, improving alignment with the unit tests through MAC network updates ensures better adherence to the actual task requirements.

**Coding team for feed-forward.** In the feed-forward process, the coding team synthesizes code according to the given coding task. To handle the extensive software requirements, the coding team is implemented as a MAC network. It divides the comprehensive requirements into a sequence of smaller, more specific function implementation subtasks, and progressively conquers them through the agentic workflow. Unlike existing MAC systems that heuristically decompose coding tasks and define the agentic workflow, we initialize the MAC network using a novel self-organizing approach. A coding organizer agent automatically and flexibly decomposes the task requirements into subtasks and assembles the coding agent team accordingly. The number of coding agents is dynamic, adjusting in response to the task requirements. Note that, the quality of the generated code is unknown during the forward pass, which necessitates the self-evolving paradigm to iteratively refine the generation.

**Testing team and compiler for feedback collection.** To verify whether the generated code meets the requirements of the coding task, we employ unit tests as the target proxy. These test cases consist of input-output pairs tailored to specific requirements. For example, a test case for a keyboard control requirement would detail the type of control as the input and describe the expected behavior as the output. To create flexible and comprehensive unit tests, we set up the testing team as a MAC network and also initialize it in a self-organized way. A testing organizer agent automatically decomposes our specified key testing criteria into subtasks and accordingly forms the testing agent team .

Once the test cases and generated code are ready, they are executed in the compiler, which functions as the environmental tool, producing execution logs. These logs clearly point out the gap between the generated code and the test cases. It shows satisfied testing requirements, existing function errors, and unmet testing requirements. This feedback information can be used to verify whether each agent's subtask is accomplished and guide the MAC network update.

**Updating team for textual back-propagation.** The updating team consists of two collaborative agents: the gradient agent and the updating agent, adjusting the MAC network based on the execution logs, including the agent prompts and workflows. This process consists of two steps. First, the gradient agent summarizes the textual gradient by identifying accomplished subtasks for satisfied requirements, appending new subtasks for unmet requirements, and analyzing errors to detail their originating agents and revising suggestions. Second, the updating agent modifies the coding agent team by removing agents that have completed their subtasks, adding new agents for the new subtasks, and revising agent prompts to address issues identified in the previous generation. The agent workflow is updated once the agent team is revised, based on the dependencies among the subtasks.

## 4    rSDE-Bench: requirement-Oriented Software Development Engineering Benchmark

This section introduces rSDE-Bench, a requirement-oriented benchmark designed to assess the ability of models to handle software-level coding tasks. Each coding task involves multiple detailed software requirements. These requirements specify each functionality and constraint of the software, item by item, serving as measurable benchmarks for assessing the software's effectiveness. As shown in Fig. 3, unlike previous instruction-oriented approaches Qian et al. (2023); Hong et al. (2023) which

rely on brief instructions as input, `rSDE-Bench` uses comprehensive software requirements as input, complemented by unit test cases to automatically evaluate the correctness. This benchmark provides software-level coding tasks and automatic evaluation, aligning more closely with real-world software development practices.

## 4.1 BENCHMARK CONSTRUCTION

rSDE-Bench involves two typical real-world software types: game and website. They can reflect different coding capacities demanded in realistic software development. Game often requires handling dynamic interactions, real-time state changes, and user-driven operations, focusing on elements like logic execution, initialization, and game state transitions. Website emphasizes static and dynamic content management, user interaction through forms and buttons, and ensuring page elements are displayed and functional.

rSDE-Bench involves diverse requirements, each paired with a test case. Specifically, rSDE-Bench provides 53 unique coding tasks and 616 test cases. rSDE-Bench introduces two requirement difficulty levels, including basic and advanced, to reflect the varying complexity of real-world software development tasks. For details on the benchmark construction, software statistics, software requirements, and test case examples, see Sec. 7 in the Appendix.

## 4.2 AUTOMATIC EVALUATION

rSDE-Bench supports automatic evaluation of requirement correctness. It achieves this by pairing a specifically designed black-box test case with each requirement. The test case can directly verify whether the generated code achieved the requirement. Its evaluation metric is the accuracy, which quantifies the proportion of correctly passed test cases. It is similar to the `pass@1` metric in HumanEval Chen et al. (2021), which evaluates the pass ratio of correctly achieved functions against the total functions via unit test verification. It is a fully automated evaluation process, eliminating the need for human involvement while still providing accurate and reliable assessments.

Previous benchmarks for software code generation mainly rely on two evaluation methods. One method is human evaluation Hong et al. (2023), which is time-consuming and not scalable for large datasets. The other method is indirect evaluations Qian et al. (2023), which defines metrics like consistency, completeness, and quality. Consistency measures how closely the generated software aligns with the original requirement description by comparing the cosine similarity between the two. Completeness is determined by detecting the presence of placeholder (such as `pass` or `TODO`), which results in a binary value of 0 or 1. Quality is then calculated as the product of several factors: consistency, completeness, and executability. As illustrated in Fig. 3, they could not measure the correctness of the generated code in fulfilling requirements. In contrast, `rSDE-Bench`'s test cases-based evaluation is more rigorous and precise. These test cases can accurately verify the correctness of generated code in fulfilling the requirements. `rSDE-Bench` promises reliable and scalable automatic evaluation. We have validated the significant advantages of the proposed automatic evaluation over the previous metrics, including consistency and quality; see Fig. 4.

## 5 EXPERIMENTS

### 5.1 EXPERIMENTAL SETUP

**Baselines.** To validate the effectiveness of our EvoMAC, we conducted comparisons against both single-agent and multi-agent baselines. The single-agent baselines involve three prominent large models: GPT-4o-Mini (gpt-4o-mini), Claude-3.5-Sonnet (claude-3-5-sonnet-20240620), and Gemini (gemini-1.5-flash). For multi-agent baselines, we included five state-of-the-art (SOTA) methods: MetaGPT Hong et al. (2023), Autogen Wu et al. (2023), Mapcoder Islam et al. (2024), Agentverse Chen et al. (2023), and ChatDev Qian et al. (2023). To ensure a fair comparison, all multi-agent baselines, including our EvoMAC, are powered by the efficient and powerful GPT-4o-Mini model. Additionally, to demonstrate the adaptability and robustness of our EvoMAC, we developed two EvoMAC variants using GPT-4o-Mini and Claude-3.5-Sonnet.

**Datasets.** Our experiments cover both the proposed rSDE-Bench and the standard coding benchmark HumanEval Chen et al. (2021). HumanEval comprises 164 Python function completion problems, where the task is to generate code from a single function description.

### 5.2 EFFECTIVENESS OF RSDE-BENCH'S EVALUATION AND EVOMAC

**rSDE-Bench's automatic evaluation metric (accuracy) is highly aligned with human evaluation.** Our primary goal is to validate the effectiveness of the proposed automatic evaluation in rSDE-Bench

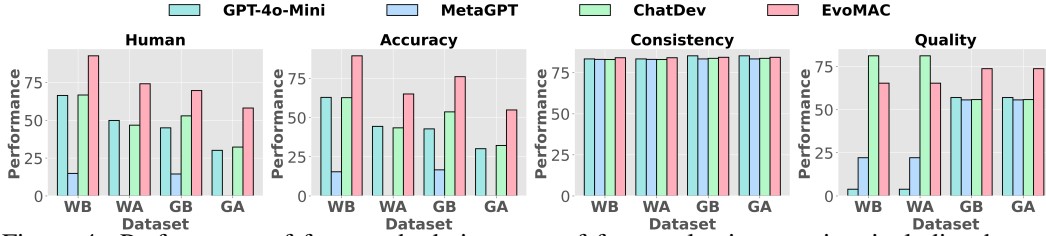

Figure 4: Performance of four methods in terms of four evaluation metrics, including human evaluation, our automatic evaluation (accuracy), consistency, and quality. WB/GB and WA/GA represent Web/Game Basic and Web/Game Advanced respectively. Our accuracy metric is highly aligned with human evaluation across four dataset settings.

Table 1: Comparison of EvoMAC with five multi-agent and three single-agent SOTA baselines, all powered by GPT-4o-Mini. **Red** values represent the percentage improvement of EvoMAC, shade in pink, over the single-agent baselines, shade in grey.

| Method | Model | rSDE-Bench | | | | HumanEval |
|---|---|---|---|---|---|---|
| | | Website(%) | | Game(%) | | (%) |
| | | Basic | Advanced | Basic | Advanced | Pass@1 |
| Single-Agent | Gemini-1.5-Flash | 29.79±1.00 | 11.61±2.34 | 21.74±6.39 | 6.45±6.97 | 73.17 |
| | Claude-3.5-Sonnet | 58.90±1.48 | 37.11±1.06 | 44.20±5.41 | 18.29±13.26 | 89.02 |
| | GPT-4o-Mini | 62.90±2.52 | 44.40±4.21 | 42.76±15.50 | 30.10±11.87 | 88.41 |
| Multi-Agent | MetaGPT | 15.41±0.00 | 0.00±0.00 | 16.67±2.71 | 0.00±0.00 | 88.41 |
| | Autogen | 25.68±4.14 | 5.40±3.34 | 17.39±1.78 | 0.00±0.00 | 85.36 |
| | MapCoder | 34.70±1.59 | 14.57±0.66 | 29.71±6.72 | 7.52±6.10 | 90.85 |
| | Agentverse | 15.41±0.00 | 0.00±0.00 | 37.67±8.20 | 16.13±4.55 | 90.85 |
| | ChatDev | 62.67±0.28 | 43.45±0.77 | 53.63±5.70 | 32.26±4.55 | 70.73 |
| | EvoMAC | **89.38±1.01** | **65.05±1.56** | **77.54±2.04** | **51.60±4.54** | **94.51** |
| | | **+26.48** | **+20.65** | **+34.78** | **+21.50** | **+6.10** |

by comparing it with two existing evaluation metrics: consistency and quality, both from SRDD Qian et al. (2023). For a fair comparison, our golden standard is human evaluation, conducted by two expert code engineers who manually verify the fulfillment of requirements by interacting with the developed software. This process is tedious, taking around four hours per expert to evaluate the entire benchmark. The effectiveness of an evaluation metric depends on how closely it aligns with human evaluation.

Fig. 4 presents the performance of four methods in terms of four evaluation metrics, including human evaluation, our automatic evaluation, consistency, and quality. We see that: i) our automatic evaluation is highly aligned with human evaluation across two software types (Website and Game), four methods, (GPT-4o-Mini, MetaGPT, ChatDev, and our EvoMAC), and two requirement difficulties (Basic and Advanced). The correlation coefficient between human evaluation and our accuracy metric is 0.9922, demonstrating the effectiveness of the proposed automatic evaluation in rSDE-Bench; ii) Consistency and quality metrics differ significantly from human evaluation, with correlation coefficients of 0.2583 and 0.3041, respectively. This discrepancy occurs because consistency in SRDD measures similarity, and quality in SRDD focuses on executability, which does not guarantee that all requirements are met. This highlights the need for rSDE-Bench, as the SRDD benchmark does not support requirement-oriented software development.

**EvoMAC outperforms previous SOTAs on both software-level and function-level coding benchmarks: rSDE-Bench and HumanEval.** Tab. 1 compares EvoMAC with five multi-agent and three single-agent SOTA baselines, all powered by GPT-4o-Mini for a fair comparison. We see that EvoMAC significantly outperforms previous SOTAs across all datasets. EvoMAC outperforms single-agent methods by 26.48% on the rSDE-Bench Website Basic and 34.78% on the rSDE-Bench Game Basic, as well as surpassing existing multi-agent methods by over 20%. This highlights the effectiveness of multi-agent collaboration and the power of EvoMAC.

### 5.3 EFFECTIVENESS OF EVOLVING

Fig. 6 shows the accuracy of EvoMAC over multiple evolving iterations on the rSDE-Bench and HumanEval. Each figure presents two curves: one for EvoMAC powered by GPT-4o-Mini (red) and the other by Claude-3.5 (blue). We have the following findings:

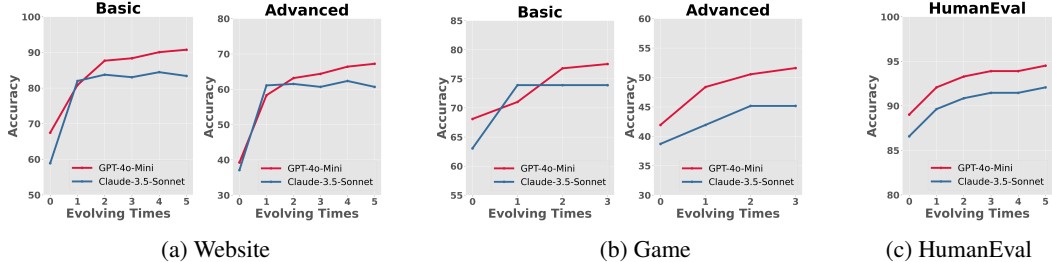

(a) Website        (b) Game        (c) HumanEval

Figure 6: Effect of EvoMAC performance across evolving times empowered by GPT-4o-Mini and Claude-3.5-Sonnet on Website, Game, and HumanEval datasets. The figure shows EvoMAC continuously improves with the evolving times on both LLM drives.

Table 2: Ablation study about coding/testing team with single/multi-agent, with/without evolving, and with/without environment tool. Best performances are bolded.

| | Coding | Testing | Evol. | Env. | Website(%) Basic | Website(%) Advanced | Game(%) Basic | Game(%) Advanced |
|---|---|---|---|---|---|---|---|---|
| a) | Single | - | - | - | 63.70 | 41.70 | 42.76 | 30.10 |
| b) | Multi | - | - | - | 67.47 | 39.27 | 68.10 | 41.93 |
| c) | Single | Single | ✓ | ✓ | 80.82 | 60.32 | 71.73 | 41.93 |
| d) | Multi | Single | ✓ | ✓ | 83.90 | 60.72 | 76.08 | 41.93 |
| e) | Single | Multi | ✓ | ✓ | 83.56 | 61.94 | 73.91 | 45.16 |
| f) | Multi | Multi | ✓ | - | 78.08 | 52.23 | 55.80 | 33.32 |
| g) | Multi | Multi | ✓ | ✓ | **90.75** | **67.20** | **77.54** | **51.60** |

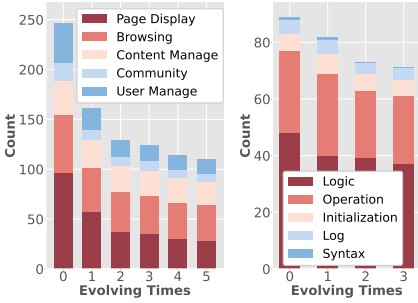

Figure 5: Failure case distribution across evolving times on Website and Game.

**EvoMAC continuously improves with the evolving times.** Fig. 6 shows that as evolving iterations increase, performance consistently improves across all five dataset settings, covering two difficulty levels, two software types, and both requirement-oriented and function complement benchmarks. This highlights the effectiveness, generalizability, and robustness of the self-evolving approach, encouraging EvoMAC to evolve whenever possible.

**EvoMAC indistinguishably improves with different driving LLM.** From Fig. 6, we see that: i) both EvoMAC variants continuously improve with evolving iterations, demonstrating the robustness of the self-evolving design; ii) the two curves do not intersect, indicating that the EvoMAC variant powered by a more powerful single model consistently outperforms the other, highlighting the advantage of using a stronger model. Success builds on success.

**Failure case analysis.** Fig. 5 shows the failure case statistics across iterations for Website and Game, showing a general decrease in errors as iterations progress. We see that: i) the most common errors are page display issues in Website and logic errors in Game; ii) page errors are resolved more quickly, while logic errors persist, suggesting that more isolated issues are easier to fix during the evolution process. This results in a sharp initial performance improvement as sipler problems are addressed early, followed by a plateau as more complex issues remain unresolved, shown in Fig. 6.

## 5.4 ABLATION STUDY

To assess the effectiveness of each component, Tab. 2 details an ablation study featuring seven EvoMAC variants.

**Effectiveness of objective environment feedback.** Environment feedback, such as code execution logs, is essential for software development. Variant f) omits this tool, instead using an LLM-driven agent to critique the code. Comparing Variant g) with Variant f) shows a notable performance drop: Website tasks decrease by 12.67% and 14.97%, and Game tasks by 21.74% and 18.28% for Basic and Advanced levels, respectively. This underscores the importance of objective environmental feedback, as agent-driven critiques may introduce bias and fail to guide the evolution effectively.

**Effectiveness of multi-agent collaboration in coding team and testing team.** Comparing Variant g) to Variant e), we observe a performance decrease of 7.19% and 5.26% on Website Basic and Advanced respectively, when the coding team is reduced to a single agent. Similarly, comparing Variant g) to Variant d), there is a performance drop of 6.85% and 6.48% on Website Basic and Advanced respectively, also when the team is reduced to a single agent. These results demonstrate

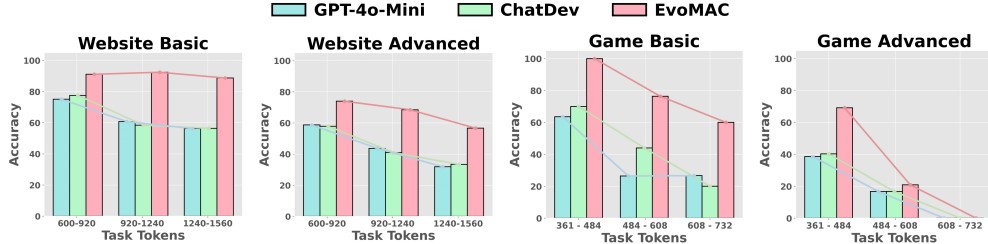

Figure 7: EvoMAC outperforms previous multi-agent and single-agent systems across all the context lengths across the four dataset settings on rSDE-Bench.

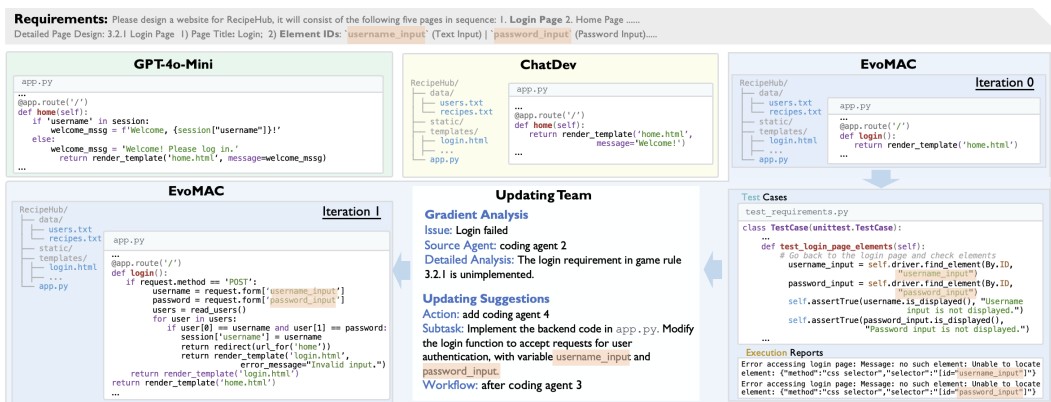

Figure 8: We show the generated code of single-agent, GPT-4o-Mini, and multi-agent systems, ChatDev, and our EvoMAC (iteration =0/1) given the Website task (RecipeHub). After evolving, EvoMAC can revise previous issues and fulfill the task requirement.

the necessary for involving multi-agent collaboration, highlighting that multi-agent setups offer more flexible adjustments and enhanced capabilities for evolution.

**Effectiveness in handling varied task token lengths.** Fig. 7 shows a comparison of task token lengths and performance across GPT-4o-Mini, ChatDev, and EvoMAC. We see that: i) EvoMAC consistently outperforms ChatDev and GPT-4o-Mini across all context lengths, with its self-evolving mechanism enabling the identification and correction of missed contexts and errors during iterations; ii) EvoMAC experiences less performance degradation on the rSDE-Bench Website than on the Game, as Website tasks are more modular and can be broken into subtasks, whereas Game tasks require more coordinated management, making them more challenging.

## 5.5 CASE STUDY

Fig. 8 presents the generated code by a single agent, GPT-4o-Mini, multi-agent systems, ChatDev, and our EvoMAC before and after evolving (iteration=0/1). We see that: i) EvoMAC after evolving can correct issues from previous iterations and successfully fulfill the task requirements; ii) multi-agent systems tend to better comprehend the task requirements and produce more well-structured code. More generated software can refer to Sec. 10 in the Appendix.

## 6 CONCLUSION

We propose EvoMAC, a novel self-evolving paradigm for MAC networks. EvoMAC iteratively adapts agents and their connections during the testing phase of each task. It achieves this with a novel textual back-propagation algorithm. EvoMAC can push coding capabilities beyond function-level tasks and into more complex, software-level development. Furthermore, we propose rSDE-Bench, a novel requirement-oriented software development benchmark. rSDE-Bench features both complex and diverse software requirements, as well as the automatic evaluation of requirement correctness. Comprehensive experiments validate that the automatic requirement-aware evaluation in rSDE-Bench aligns closely with human evaluation. EvoMAC outperforms previous SOTAs in both software-level rSDE-Bench and function-level HumanEval benchmarks.

**Future works.** In the future, we plan to introduce a reward model to enhance the self-evolving paradigm's ability to learn from feedback and extend the rSDE-Bench to more software types.

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

APPENDIX

# 7 BENCHMARK DETAILS

Table 3: Basic statistics for website and game domains, including the amount of samples, prompt length (mean/max), and number of test cases at both Basic and Advanced levels.

| Benchmark | Software | | Test Case | |
|---|---|---|---|---|
| | Amount | Length | Basic | Advanced |
| Website | 45 | 1011/1553 | 292 | 247 |
| Game | 8 | 507/788 | 46 | 31 |

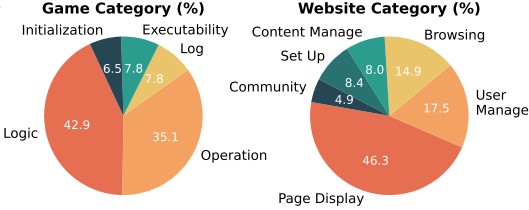

Figure 9: Statistics of Game and Website tasks.

## 7.1 FEATURES

**Challenging and diverse software requirements.** rSDE-Bench features long-context software requirements (averaging 507/1011 words for game and website tasks, respectively), unlike instruction-oriented benchmarks Chen et al. (2021); Austin et al. (2021); Jimenez et al. (2023) that rely on brief prompts. These detailed requirements better reflect real-world lengthy and complex software development challenges.

**Requirement-aware precise and efficient evaluation.** rSDE-Bench employs detailed software requirements and automated unit tests to precisely measure how well generated software meets its objectives. Generated codes are evaluated based on pass rates from running specific test cases, offering an accurate and efficient process. In contrast, instruction-oriented benchmarks rely on brief prompts, which lack constraints and make evaluation less reliable, often requiring labor-intensive or indirect evaluation.

## 7.2 CONSTRUCTION PROCESS

**Step 1: Software requirement generation.** Each task instance begins with the generation of clear, measurable software requirements. Given the inherent differences across various types of software, we adopt distinct approaches for their formulation. For game-related software, we focuses on common real-world games, capturing detailed task requirements such as GUI layout initialization, interaction methods, and game rules. To align more closely with actual game development practices, we also include game state logging as part of the software requirements. Due to the complexity of logic in game software, these requirements are manually crafted by human. In contrast, for website-related software, we begin with a concise website name, and then leverage the large language model (gpt-4o-mini) to enrich the requirements according to predefined patterns. This approach ensures both efficiency and scalability in the creation of benchmarks for websites. By tailoring the process to the distinct characteristics of each software domain, we maintain precision in requirement formulation while addressing the unique challenges posed by each context.

**Step 2: Requirement-based test cases generation.**

As illustrated in Fig 10 and Fig 11, unit tests offer a precise evaluation of software completion. Each task instance includes black-box unit test cases that correspond directly to the software requirements, allowing for a quantitative assessment of requirement fulfillment. To further assess the model's code generation capabilities, we categorize test cases into two levels of difficulty—basic and advanced, as outlined in Tab. 3. We also provide an overview of all websites and games in Tab. 4 and Tab. 5 respectively. As shown in Fig. 9, test cases for website and game software exhibit structural differences, reflecting the distinct nature of each software type. They enable more targeted evaluation of code generation capabilities. Thus, similar to software requirements, the test cases are constructed differently based on the software type. For game-related tests, we manually create test cases, akin to the HumanEval Chen et al. (2021) benchmark, which tracks state changes in response to specific inputs. In the game environment, we assess how game states evolve in response to GUI interactions. For website-related tests, large language model (gpt-4o-mini) generates Selenium-based test cases aligned with the software requirements, followed by manual corrections to resolve any ambiguities. This structured approach ensures rigorous evaluation across diverse software domains.

Table 4: Overview of Websites in rSDE-Bench.

| Websites | | |
|---|---|---|
| CharitableGivingPlatform | DailyHealthTips | DailyJournalApp |
| EcoFriendlyLivingTips | ElderCareResources | EventPlanner |
| FitnessEquipmentRental | FitnessTracker | FreelancerMarketplace |
| GreenLivingGuide | HealthConsultationPlatform | MotivationalQuotesApp |
| MusicFestivalDirectory | NoteTakingApp | NutritionInformationHub |
| OnlineLibraryManagementSystem | OnlineTherapeuticJournaling | OnlineThriftStore |
| PeerTutoringNetwork | PersonalBlog | PersonalFinanceBlog |
| RecipeHub | RemoteInternshipMarketplace | RemoteJobBoard |
| TravelDiary | VirtualBookPublishing | VirtualWellnessRetreats |
| DigitalArtworkGallery | DigitalStorytellingPlatform | ExpenseTracker |
| FitnessChallenges | GardeningForBeginners | GourmetFoodSubscription |
| MovieRecommendationSystem | MusicCollaborator | OnlineCulturalExchange |
| OnlineCulturalFestivals | OnlineVintageMarket | ParentingAdviceForum |
| PetCareCommunity | PortfolioSite | SkillShare |
| TaskManager | VolunteerMatch | OnlineShoppingCenter |

Table 5: Overview of Games in rSDE-Bench.

| Games | | | |
|---|---|---|---|
| Balls | Tank | Racing | Ghostly |
| Mario | Bomberman | Sokoban | Brick |

**Basic and advanced requirements definition.** The basics reflect the fundamental and more achievable requirements, such as interaction, control, and logging. The advanced reflects more complex software functionalities, such as game logical rules, and dynamic web content management. For the games, basic requirements involve straightforward user interactions that do not require complex logic, such as character movement or interacting with simple GUI elements. Advanced requirements incorporate more intricate logic, such as managing game state transitions based on user actions or handling conditional game events. These cases focus on ensuring the correct execution of basic actions. In contrast, advanced cases incorporate more intricate logic, such as managing game state transitions based on user actions or handling conditional game events. These cases challenge the model's ability to generate code that integrates dynamic decision-making and interaction within the game environment. For websites, basic cases focus on ensuring that the necessary page elements—such as input fields, buttons, and layouts—are present correctly. These cases assess the completeness of the webpage's structure. On the other hand, advanced cases evaluate more complex functionality, such as handling user authentication, managing dynamic content, or executing specific operations within a content management system. These cases require the model to generate code that performs backend logic and manages user interactions at a deeper level.

## 8 ALGORITHM

In this section, we present the algorithm of EvoMAC in Alg. 1. For more details, please refer to Section 3.

## 9 CASE STUDY

### 9.1 COMPLETE EVOMAC PROCESS

In this section, we show a complete process of EvoMAC on RSD-Bench. Please refer to Tab. 6.

**Software description**

```
Task: Develop a simple Sokoban game. You must design a GUI.
Requirements:
1. The game board should be divided into grid squares.
2. Players will control the game using the arrow keys on the keyboard.
3. As the game starts, a log file named 'game.log' should be created to record the game's progress. The content of the
game.log file should be appended with a new entry after each player action.The content of the game.log file should be
cleared (if any) at the start of each game session.
Each log entry should follow this format:
{
"timestamp": timestamp,
"EVENT_TYPE": "MOVE_RIGHT" | "MOVE_LEFT" | "MOVE_UP" | "MOVE_DOWN" | "INVALID_MOVE",
"player_position": [x, y],
"box_positions": [[x1, y1], [x2, y2], ...],
"game_status": "ONGOING" | "COMPLETE"
}
4. The victory conditions for the game is: All boxes are pushed onto their corresponding coordinate point.
5. The initial positions of each element are required as follows:
player_position = [1, 1]
box_positions = [[3, 3], [4, 2]]
goal_positions = [[5, 5], [6, 3]]
([3, 3] is the initial position of the first box whose target position is [5, 5].  [4, 2] is the initial position of
the second box whose target position is [6, 3].)
wall_positions = [[0, 4], [1, 4], [2, 4],[3, 4],[4, 4]]
(the first numnber in each pair is the x-coordinate and the second number is the y-coordinate)
```

**Evaluation functions**

```
check_Excutablity  check_log  check_move_right  check_move_left  check_move_box
check_move_wall  check_seqbox  check_end  check_wrong_end
```

Figure 10: Test cases of Game in rSDE-Bench.

**Software description**

```
# Requirement Document for DailyHealthTips Web Application

## 1. Objective
Develop a web application named 'DailyHealthTips' that provides users with daily health tips, allowing them to receive
advice and information about maintaining a healthy lifestyle, using Python as the development language. Note that the
website should start from the login page.

## 2. Language
The required development language for the DailyHealthTips web application is Python.

## 3. Page Design

### Page 1: Login Page
- **Page Title**: User Login
- **Overview**: This page allows users to log in to their accounts.
- **Elements**:
  - **Username Field**:
    - **ID**: `username_field`
  - **Password Field**:
    - **ID**: `password_field`
  - **Login Button**:
    - **ID**: `login_button`
…………
```

**Evaluation functions**

```
test_login_page_elements  test_login_page_functionality  test_daily_tips_page_elements
test_daily_tips_page_functionality  test_tips_archive_page_elements  test_tips_archive_page_functionality
```

Figure 11: Test cases of Website in rSDE-Bench.

## 9.2   UNIT TEST CASE

In this section, we show more unit test cases written by coder on RSD-Bench, please refer to Tab. 13 and Tab. 16.

## 9.3   UPDATING PROCESS

In this section, we show additional examples of the updating process on RSD-Bench and HumanEval dataset. Please refer to Tab. 19 and Tab. 22 respectively. For RSD-Bench, due to the code length, we only show the texture updating process(codes are available at Sec. 9.1). We can see that the updating agent will adjust the job of each coder dynamically according to the result of test team.

---

**Algorithm 1** Self-Evolving Paradigm

---

**Require: X**                      ▷ Task input
**Require:** $\mathcal{A}_g^{(0)}$         ▷ Initialized MAC network: agent prompts and pipeline
**Require:** $\mathcal{A}_t$          ▷ Designed MAC network to generate target proxy
**Require:** $\mathcal{G}$                ▷ Agent-based gradient function
**Require:** $\mathcal{U}$                ▷ Agent-based update function
**Require:** $E$              ▷ Environment tool to generate loss
 1: Define $K$ as the number of self-evolving iterations, $\Phi$ as MACN generation process
 2: # Target Proxy
 3: $\mathbf{T} = \Phi(\mathbf{X}, \mathcal{A}_t)$
 4: # Self-Evolving Procedure
 5: **for** $k = 0, 1, \ldots, K - 1$ **do**
 6:    # Forward Pass
 7:    $\mathbf{G}^{(k)} = \Phi(\mathbf{X}, \mathcal{A}_g^{(k)})$
 8:    # Loss Computation
 9:    $\mathbf{L}^{(k)} = \langle \mathbf{G}^{(k)}, \mathbf{T} \rangle_E$         ▷ Use environment feedback as textual loss
10:    # Textual Backpropagation
11:    $\nabla \mathbf{L}^{(k)} = \mathcal{G}(\mathbf{L}^{(k)}, \mathcal{A}_g^{(k)})$        ▷ Summarize textual gradient
12:    $\mathcal{A}_g^{(k+1)} = \mathcal{U}(\mathcal{A}_g^{(k)}, \nabla \mathbf{L}^{(k)})$      ▷ Update agent prompts and pipeline
13: **end for**
14: **return** $\mathcal{A}_g^{(K)}, \mathbf{G}^{(K)}$

---

## 10   SOFTWARE PRESENTATION

In this section, we show some games and websites written by EvoMAC. Fig. 12 and Fig. 13 present the games and websites respectively. We see that: i) EvoMAC outputs games with well-written GUI and game rules. It can handle different kinds of GUI and game rule requirements from diverse games. ii) EvoMAC outputs websites with beautified, user-friendly web pages and correct transition logic. It can handle the requirements of different websites.

Table 6: A complete iteration process of EvoMAC on RSD-Bench

| Notation | Meaning | Example | Real example |
|---|---|---|---|
| $\mathbf{X}$ | Textual description of the task to be completed. | A coding task such as: "Implement a code that simulates keyboard input processing via Python." | See Tab. 7 |
| $\mathcal{A}_g$ | MAC network representing the team responsible for generating the code. | The coding team consists of coding agents completing subtasks in sequence. | See Tab. 8 |
| $\mathbf{G} = \Phi(\mathbf{X}, \mathcal{A}_g)$ | Generated output produced by the coding team as a result of the feed-forward pass. | The generated code: '...def process_input(keyboard_input): ...' | See Tab. 9 |
| $\mathcal{A}_t$ | MAC network representing the team responsible for generating the target proxy (unit tests). | The testing team generates unit tests for the task. | See Tab. 10 |
| $\mathbf{T} = \Phi(\mathbf{X}, \mathcal{A}_t)$ | Target proxy (unit tests) generated by the testing team based on the task description. | Unit tests like: '...def test_press_input(): assert process_input('Enter') == 'Processed Enter'' | See Tab. 11 |
| $< \mathbf{G}, \mathbf{T} >_E$ | Environmental feedback comparing the generated output $\mathbf{G}$ with the target proxy $\mathbf{T}$ using an objective environment (e.g., compiler or test results). | The environment executes the generated code against the unit tests, providing feedback like: 'Failure: test_press_input' | The execution outcome of the unit test from the terminal. If the execution is successful, the outcome is 'The software run successfully without errors.' |
| $\min < \mathbf{G}, \mathbf{T} >_E$ | The optimization objective aiming to minimize the difference between the generated output and the target proxy using the environmental feedback. | Based on feedback, the system iteratively refines the coding team to generate code that better meets the task. | See Tab. 12 |

Table 7: **Textual description of the task to be completed.**

Task: Design a Single-Player Tank Battle Game
Requirements:
1. The interface should be divided into a 20x20 grid, though grid lines are not necessary. Each tank occupies one grid space, while obstacles may occupy multiple grid spaces. The background should be black, obstacles should be brown, enemy tanks should be silver, and the player's tank should be yellow.
2. The player can control the tank's movement using the arrow keys on the keyboard, allowing for movement one grid space at a time. The 'enter' key is used to fire bullets.
3. In the game, there are two enemies fixed at a certain position on the game interface, constantly firing bullets in four directions: up, down, left, and right. Two enemies and players cannot be initialized in the same row.
4. Both the player and the enemies have their own health points, which are initialized to 200. When hit by a bullet, the player's health decreases by 10 and the enemy's health decreases by 100. When health points drop to zero, the corresponding tank is destroyed. But the log still records information about the destroyed tank, with health points of 0.
5. Destroying an enemy tank earns the player 200 points. The game ends when the player's tank is destroyed or all enemy tanks are destroyed, at which point the player's score will be displayed on the screen.
6. As the game start, a new log file named 'game.log' should be created to record the game's progress. The first log entry should capture the initial state of the game. Each time a new event occurs to the player, a new log entry should be written in real-time. The logs should follow the format below. The EVENT_TYPE can only be one of the following: "INIT", "MOVE_LEFT", "MOVE_RIGHT", "MOVE_UP", "MOVE_DOWN", "FIRE" or "INJURED". The game_state should capture the current state of the game. In this setup, the position coordinates are defined with the top-left grid as [0, 0], where the x-coordinate increases by one unit for each grid space moved to the right, and the y-coordinate increases by one unit for each grid space moved downward. The player is initialized at position [0,0].

```
{
    "timestamp": timestamp,
    "EVENT_TYPE": "INIT" | "MOVE_LEFT" | "MOVE_RIGHT" | "MOVE_UP"
    | "MOVE_DOWN" | "FIRE" | "INJURED",
    "game_state": {
        "player": {
            "position": [xp, yp],
            "health": health_player,
            "score": score
        },
        "enemies": [
            {
                "position": [xe1, ye1],
                "health": health_enemy1
            },
            {
                "position": [xe2, ye2],
                "health": health_enemy2
            }
        ],
        "obstacle_position": [(xo1, yo1), ..., (xon, yon)]
    },
}
```

Table 8: **MAC network representing the team responsible for generating the code.**

```
{
    "composition": {
        "Programmer 1": "Implement bullet firing logic that decreases
        health points for both the player and enemies when hit. This
        includes creating a function to handle bullet movement and
        collision detection, updating health points accordingly, and
        logging the events in the game.log file.",
        "Programmer 2": "Implement health management to check for
        health points dropping to zero and handle tank destruction.
        This includes creating a function to check health status
        after each hit, updating the game state, and logging the
        destruction event in the game.log file.",
        "Programmer 3": "Implement scoring logic to update the player's
        score when an enemy tank is destroyed and display the final
        score when the game ends. This includes creating a function
        to manage scoring and game-over conditions, and logging the
        final score in the game.log file."
    },
    "workflow": {
        "Programmer 1": [],
        "Programmer 2": [
            "Programmer 1"
        ],
        "Programmer 3": [
            "Programmer 2"
        ]
    }
}
```

Table 9: **Generated output produced by the coding team as a result of the feed-forward pass. Due to the limit of page length, we only show partial of the codes.**

```python
'''
This is the main file for the Single-Player Tank Battle Game.
It initializes the game, sets up the GUI, and manages the game
loop, including player movement, enemy behavior, and logging
events.
'''
import pygame
import json
import time
# Constants
GRID_SIZE = 20
CELL_SIZE = 30
WINDOW_SIZE = GRID_SIZE * CELL_SIZE
PLAYER_COLOR = (255, 255, 0)  # Yellow
ENEMY_COLOR = (192, 192, 192)  # Silver
OBSTACLE_COLOR = (139, 69, 19)  # Brown
BACKGROUND_COLOR = (0, 0, 0)  # Black
FPS = 60
# Game state
player_position = [0, 0]
player_health = 200
player_score = 0
enemies = [
    {"position": [5, 5], "health": 200},
    {"position": [10, 10], "health": 200}
]
obstacles = [(3, 3), (7, 7), (12, 12)]
# Log file
log_file = open('game.log', 'w')
initial_state = {
    "timestamp": time.time(),
    "EVENT_TYPE": "INIT",
    "game_state": {
        "player": {
            "position": player_position,
            "health": player_health,
            "score": player_score
        },
        "enemies": enemies,
        "obstacle_position": obstacles
    }
}
log_file.write(json.dumps(initial_state) + '\n')
# Initialize Pygame
pygame.init()
screen = pygame.display.set_mode((WINDOW_SIZE, WINDOW_SIZE))
pygame.display.set_caption("Tank Battle Game")
clock = pygame.time.Clock()
class Bullet:
    '''
    This class represents a bullet fired by the player.
    '''
    def __init__(self, position, direction):
        self.position = position
        self.direction = direction
    def move(self):
        if self.direction == 'UP':
            self.position[1] -= 1
        elif self.direction == 'DOWN':
            self.position[1] += 1
        elif self.direction == 'LEFT':
            self.position[0] -= 1
        elif self.direction == 'RIGHT':
            self.position[0] += 1
```
21

Table 10: **MAC network representing the team responsible for generating the target proxy (unit tests).**

```
{
    "composition": {
        "Programmer 1": "Carefully test the logging mechanism
        according to the task requirements. Ensure that logs
        happen immediately after an action is taken, recording
        the most recent state. Verify the logging order, ensuring
        basic operations are recorded first, followed by subsequent
        events. Check that the data format, keys, and values are
        accurate, paying attention to the nested data types and
        carefully checking each element.",
        "Programmer 2": "Test the logging mechanism for special
        triggered conditions, such as when the player's health drops
        to zero or when an enemy tank is destroyed. Ensure that these
        events are logged correctly and that the game state reflects
        the changes accurately.",
        "Programmer 3": "Test the value initialization required by
        the task to ensure they are correctly achieved. Pay attention
        to the coordinates of the player and enemies, ensuring they
        are initialized in the correct positions and that no two tanks
        occupy the same row.",
        "Programmer 4": "Test the function inputs and the global
        variables imported in each function. Ensure that the input
        values and global variables used in the functions are valid
        and involved when the function is called.",
        "Programmer 5": "Test that each event in the task is
        implemented correctly and that the logic triggered matches
        the conditions described in the task. This includes movement,
        firing bullets, and health reduction for both the player
        and enemy tanks."
    },
    "workflow": {
        "Programmer 1": [],
        "Programmer 2": [
            "Programmer 1"
        ],
        "Programmer 3": [
            "Programmer 1",
            "Programmer 2"
        ],
        "Programmer 4": [
            "Programmer 1",
            "Programmer 2",
            "Programmer 3"
        ],
        "Programmer 5": [
            "Programmer 1",
            "Programmer 2",
            "Programmer 3",
            "Programmer 4"
        ]
    }
}
```

Table 11: **Target proxy (unit tests) generated by the testing team based on the task description. Due to the limit of page length, we only show one test case written by one of the five test coders.**

```python
import unittest
import json
import time
from main import player_position, player_health, player_score,
enemies, log_file, log_event, fire_bullet, check_health
class TestTankBattleGame(unittest.TestCase):
    def setUp(self):
        global player_position, player_health, player_score, enemies
        player_position = [0, 0]
        player_health = 200
        player_score = 0
        enemies = [
            {"position": [5, 5], "health": 200},
            {"position": [10, 10], "health": 200}
        ]
        log_file = open('game.log', 'w')
    def tearDown(self):
        log_file.close()
    def test_move_left(self):
        global player_position
        player_position[0] = 1  # Move to the right first
        log_event("MOVE_LEFT")
        player_position[0] -= 1
        self.assertEqual(player_position, [0, 0])
    def test_move_right(self):
        global player_position
        log_event("MOVE_RIGHT")
        player_position[0] += 1
        self.assertEqual(player_position, [1, 0])
    def test_move_up(self):
        global player_position
        player_position[1] = 1  # Move down first
        log_event("MOVE_UP")
        player_position[1] -= 1
        self.assertEqual(player_position, [0, 0])
    def test_move_down(self):
        global player_position
        log_event("MOVE_DOWN")
        player_position[1] += 1
        self.assertEqual(player_position, [0, 1])
    def test_fire_bullet_hit_enemy(self):
        global player_score
        initial_health = enemies[0]["health"]
        fire_bullet()
        self.assertEqual(enemies[0]["health"], initial_health - 100)
        self.assertEqual(player_score, 200)
    def test_fire_bullet_miss_enemy(self):
        global player_position
        player_position = [0, 0]
        fire_bullet()
        self.assertEqual(enemies[0]["health"], 200)
    def test_player_injury(self):
        global player_health
        player_health -= 10
        log_event("INJURED")
        check_health()
        self.assertEqual(player_health, 190)
    def test_enemy_destruction(self):
        global enemies
        enemies[0]["health"] = 0
        check_health()
        self.assertEqual(enemies[0]["health"], 0)
if __name__ == '__main__':
    unittest.main()
```

Table 12: **The optimization objective aiming to minimize the difference between the generated output and the target proxy using the environmental feedback. According to the unit test results, Updating agent add more notes for the sub-task for Programmer 2 and Programmer 3. To see a complete updating process, please refer to Sec. 9.3**

```
{
    "composition": {
        "Programmer 1": "Implement bullet firing logic that decreases
        health points for both the player and enemies when hit. This
        includes creating a function to handle bullet movement and
        collision detection, updating health points accordingly, and
        logging the events in the game.log file.",
        "Programmer 2": "Implement health management to check for
        health points dropping to zero and handle tank destruction.
        This includes creating a function to check health status after
        each hit, updating the game state, and logging the destruction
        event in the game.log file. Additionally, ensure the game loop
        terminates properly when the player's tank is destroyed.",
        "Programmer 3": "Implement scoring logic to update the player's
        score when an enemy tank is destroyed and display the final
        score when the game ends. This includes creating a function to
        manage scoring and game-over conditions, and logging the final
        score in the game.log file. Ensure the final score is
        displayed correctly when the game ends."
    },
    "workflow": {
        "Programmer 1": [],
        "Programmer 2": [
            "Programmer 1"
        ],
        "Programmer 3": [
            "Programmer 2"
        ]
    }
}
```

Table 13: Unit test case: Game

| Example 1 | Game Ghostly |
|---|---|
| Requirement(partial) | 3. If the ghost controlled by the player eats a superpellet (the big pellets), it gains the ability to eat other ghosts. |
| Subtask given by Test Organizer(partial) | "Programmer 5": "Test that each event in the task is implemented and that the logic triggered matches the conditions in the task description." |
| Generated test case(partial) | please see Tab. 14 |
| Evaluation test case(partial) | please see Tab. 15 |

Table 14: **The partial generated test case for Game Ghostly.**

```
def test_eat_other_ghost(self):
    main.superpellet_active = True  # Activate superpellet
    main.move_ghost("up")           # Move to [1, 0]
    # more move_ghost function call omitted here
    with open('game.log', 'r') as log_file:
        logs = log_file.readlines()
        last_log = json.loads(logs[-1])
        self.assertEqual(last_log["event_type"], "eat_other_ghost")
        self.assertEqual(last_log["ghost_position"], [6, 5])
        self.assertNotIn([0, 5], main.other_ghost_positions)
```

Table 15: **The partial evaluation test case for Game Ghostly.**

```
def check_eat_other(self): # Check if Character can eat others
with superpellet
    try:
        print("==============Test eat other=============")
        process = self.start_game()
        time.sleep(0.1)
        pg.press('right',presses=1,interval=0)
        time.sleep(0.1)
        # more press omitted here
        process.terminate()
        print("terminated")
        # Read the log
        logs = self.read_log()
        print(logs)
        cnt=0
        for log in logs:
            if log["event_type"]=="eat_other_ghost":
                cnt+=1
        if cnt==2:
            print("Character can eat other with superpellet\n\n")
            return 1
        print("Character can eat other with superpellet\n\n")
        return 0
    except:
        print("eat_other error")
        return 0
```

Table 16: Unit test case: Website

| Example 2 | Website DailyHealthTips |
|---|---|
| Requirement(partial) | please see Fig. 11 |
| Subtask given by Test Organizer(partial) | "Programmer 2": "Test whether the first page of the website is the login page and whether it can log in correctly based on the example data provided in the Task." |
| Generated test case(partial) | please see Tab. 17 |
| Evaluation test case(partial) | please see Tab. 18 |

Table 17: **The partial generated test case for Website DailyHealthTips.**

```
'''
Test whether the first page of the website is the login page and
whether it can log in correctly based on the example data
provided in the Task.
'''
import unittest
from selenium import webdriver
from selenium.webdriver.common.by import By
from selenium.webdriver.common.keys import Keys
import time
class TestDailyHealthTipsLogin(unittest.TestCase):
    def setUp(self):
        self.driver = webdriver.Chrome()
        self.driver.get("http://localhost:5000")
    def test_login_page(self):
        driver = self.driver
        # Check if the current page is the login page
        self.assertIn("User Login", driver.title)
        print("Login page title is correct.")
        # Locate username and password fields
        username_field = driver.find_element(By.ID, "username_field")
        password_field = driver.find_element(By.ID, "password_field")
        login_button = driver.find_element(By.ID, "login_button")
        # Input example data
        username_field.send_keys("john_doe")
        password_field.send_keys("securepassword")
        login_button.click()
        # Wait for the page to load
        time.sleep(2)
        # Check if redirected to Daily Health Tips page
        self.assertIn("Daily Health Tips", driver.title)
        print("Successfully logged in and redirected to
        Daily Health Tips page.")
    def tearDown(self):
        self.driver.quit()
if __name__ == "__main__":
    unittest.main()
```

Table 18: **The partial evaluation test case for Website DailyHealthTips.**

```python
class TestCase(unittest.TestCase):
    def setUp(self):
        """Set up the Selenium WebDriver before each test."""
        self.driver = webdriver.Chrome()
        self.driver.get("http://localhost:5000")

    def tearDown(self):
        """Tear down the WebDriver after each test."""
        self.driver.quit()

    def login(self):
        self.driver.delete_all_cookies()
        username = "john_doe"
        password = "securepassword"
        # Performing login
        self.driver.get("http://localhost:5000")
        self.driver.find_element(By.ID, "username_field")
        .send_keys(username)
        self.driver.find_element(By.ID, "password_field")
        .send_keys(password)
        self.driver.find_element(By.ID, "login_button").click()

    # ===== Login Page Tests =====
    def test_login_page_elements(self):
        # Go back to the login page and check elements
        self.assertTrue(self.driver.find_element
        (By.ID, 'username_field').is_displayed())
        self.assertTrue(self.driver.find_element
        (By.ID, 'password_field').is_displayed())
        self.assertTrue(self.driver.find_element
        (By.ID, 'login_button').is_displayed())

    def test_login_page_functionality(self):
        # Assume user login is verified by the presence
        of daily tips page elements
        self.assertIn("User Login", self.driver.title)
        username = "john_doe"
        password = "securepassword"
        self.driver.find_element(By.ID, 'username_field')
        .send_keys(username)
        self.driver.find_element(By.ID, 'password_field')
        .send_keys(password)
        self.driver.find_element(By.ID, 'login_button').click()

        # Check if we are redirected to the daily tips page
        self.assertIn("Daily Health Tips", self.driver.title)
    # more test case omitted
```

Table 19: Updating process on RSD-Bench

| RSD-Bench Game Tank | Example |
|---|---|
| Requirement(partial) | please see Tab. 7 |
| Initial Code Organizer(iteration 0) | please see Tab. 8 |
| Updating agent(iteration 1) | please see Tab. 12 |
| Updating agent(iteration 2) | please see Tab. 20 |
| Updating agent(iteration 3) | please see Tab. 21 |

Table 20: **Updating process on RSD-Bench Game Tank, iteration 2**

```
{
    "composition": {
        "Programmer 1": "Fix the health management logic to ensure
        that when the player's health drops to zero, the destruction
        event is logged correctly, and the game loop terminates
        properly. This includes modifying the `check_health` function
        to accurately log destruction events for both the player and
        enemies.",
        "Programmer 2": "Implement the game-over conditions to ensure
        that the game ends when the player's tank is destroyed or all
        enemy tanks are destroyed. This includes creating a function
        to manage the game-over state and logging the final score
        in the game.log file."
    },
    "workflow": {
        "Programmer 1": [],
        "Programmer 2": [
            "Programmer 1"
        ]
    }
}
```

Table 21: **Updating process on RSD-Bench Game Tank, iteration 3**

```
{
    "composition": {
        "Programmer 1": "Implement constant firing behavior for enemy
        tanks in four directions. This includes creating a mechanism
        for enemies to fire bullets at the player and logging the
        corresponding events.",
        "Programmer 2": "Implement the display of the player's score
        at the end of the game when the game is over. This includes
        modifying the game-over logic to ensure the final score is
        logged and displayed correctly.",
        "Programmer 3": "Fix the logging mechanism to ensure that
        the log file remains open during gameplay and logs events
        in real-time without closing prematurely. This includes
        reviewing the log_event function and ensuring it is called
        correctly throughout the game loop."
    },
    "workflow": {
        "Programmer 1": [],
        "Programmer 2": [
            "Programmer 1"
        ],
        "Programmer 3": [
            "Programmer 1",
            "Programmer 2"
        ]
    }
}
```

Table 22: Updating process on HumanEval

| HumanEval | Example |
|---|---|
| Requirement | Write a function that takes a message, and encodes in such a way that it swaps case of all letters, replaces all vowels in the message with the letter that appears 2 places ahead of that vowel in the english alphabet. Assume only letters. |
| Initial Code Organizer(iteration 0) | "Programmer 1": "Identify and implement the logic to swap the case of all letters in the message.", "Programmer 2": "Implement the logic to replace all vowels with the letter that appears 2 places ahead in the English alphabet." |
| Generated test case(iteration 0) | please see Tab. 23 |
| Main code(iteration 0) | please see Tab. 24 |
| Updating agent(iteration 1) | "Programmer 1": AssertionError: encode('test') != 'TGST'. The actual output is: 'TgST'., "Programmer 2": AssertionError: encode('This is a message') != 'tHKS KS C MGSSCGG'. The actual output is: 'tHkS kS c MgSScGg'. |
| Final code(iteration 1) | please see Tab. 25 |

Table 23: **Updating process on HumanEval, Generated test case. iteration 0**

```
'''
Write a simple test case to verify the logic of the encode function.
'''
import unittest
from encode_function import encode
class TestEncodeFunction(unittest.TestCase):
    def test_example_case_1(self):
        self.assertEqual(encode('test'), 'TGST')
    def test_example_case_2(self):
        self.assertEqual(encode('This is a message'),
        'tHKS KS C MGSSCGG')
if __name__ == '__main__':
    unittest.main()
```

Table 24: **Updating process on HumanEval, Main code, iteration 0**

```
'''
def encode(message):
    vowels = 'aeiouAEIOU'
    encoded_message = []
    for char in message:
        if char.isalpha():
            # Swap case
            if char in vowels:
                # Replace vowel with the letter 2 places ahead
                if char.islower():
                    new_char = chr(((ord(char) - ord('a') + 2) % 26)
                    + ord('a'))
                else:
                    new_char = chr(((ord(char) - ord('A') + 2) % 26)
                    + ord('A'))
                encoded_message.append(new_char)
            else:
                # Just swap case for consonants
                encoded_message.append(char.swapcase())
        else:
            # If it's not a letter, we can just append it as is
            encoded_message.append(char)
    return ''.join(encoded_message)
```

Table 25: **Updating process on HumanEval, Final code, iteration 1**

```
'''
def encode(message):
    vowels = 'aeiouAEIOU'
    encoded_message = []
    for char in message:
        if char.isalpha():
            # Swap case
            if char in vowels:
                # Replace vowel with the letter 2 places ahead
                and swap case
                if char.islower():
                    new_char = chr(((ord(char) - ord('a') + 2) % 26)
                    + ord('a')).upper()
                else:
                    new_char = chr(((ord(char) - ord('A') + 2) % 26)
                    + ord('A')).lower()
                encoded_message.append(new_char)
            else:
                # Just swap case for consonants
                encoded_message.append(char.swapcase())
        else:
            # Preserve spaces
            encoded_message.append(char)
    return ''.join(encoded_message)
```

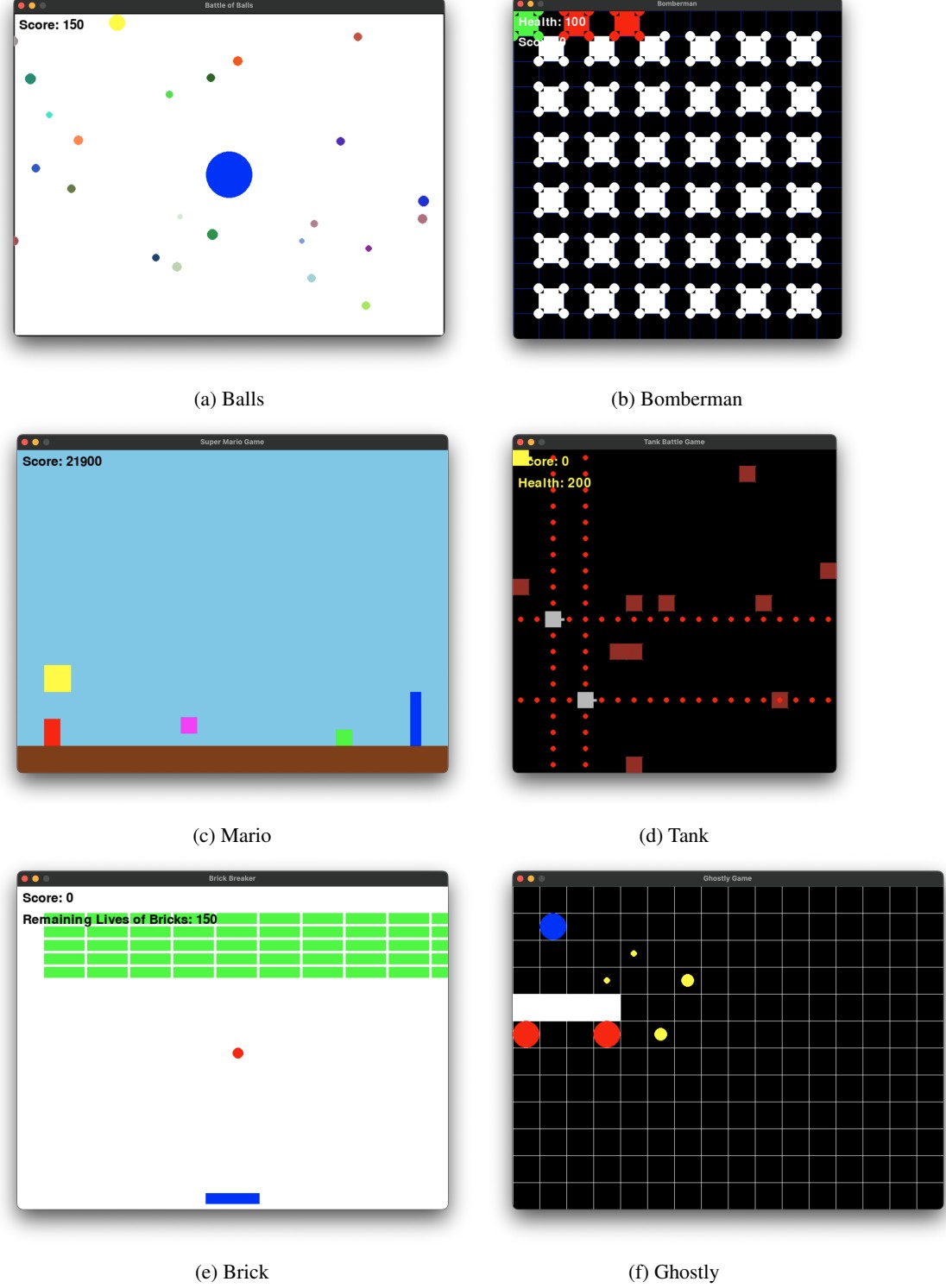

(a) Balls

(b) Bomberman

(c) Mario

(d) Tank

(e) Brick

(f) Ghostly

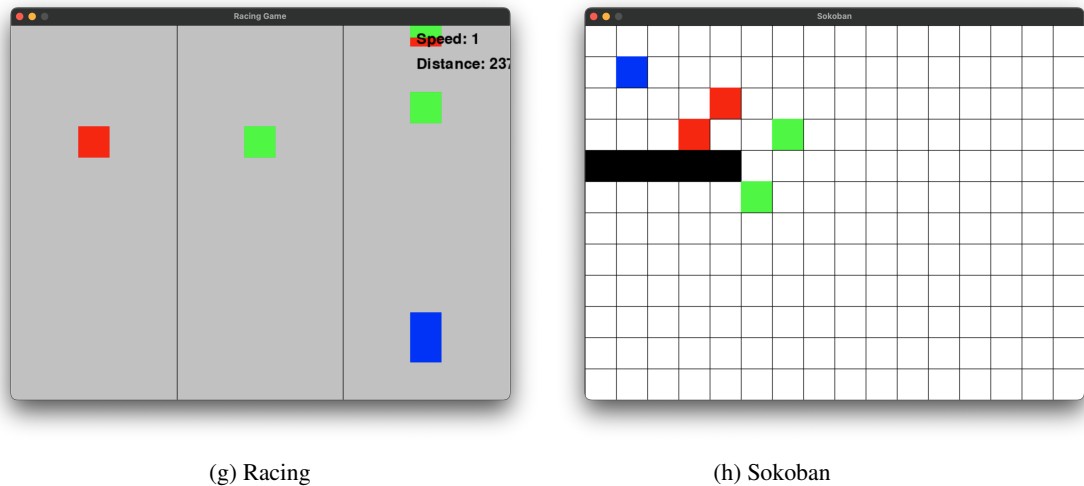

(g) Racing                  (h) Sokoban

Figure 12: Games generated by EvoMAC.

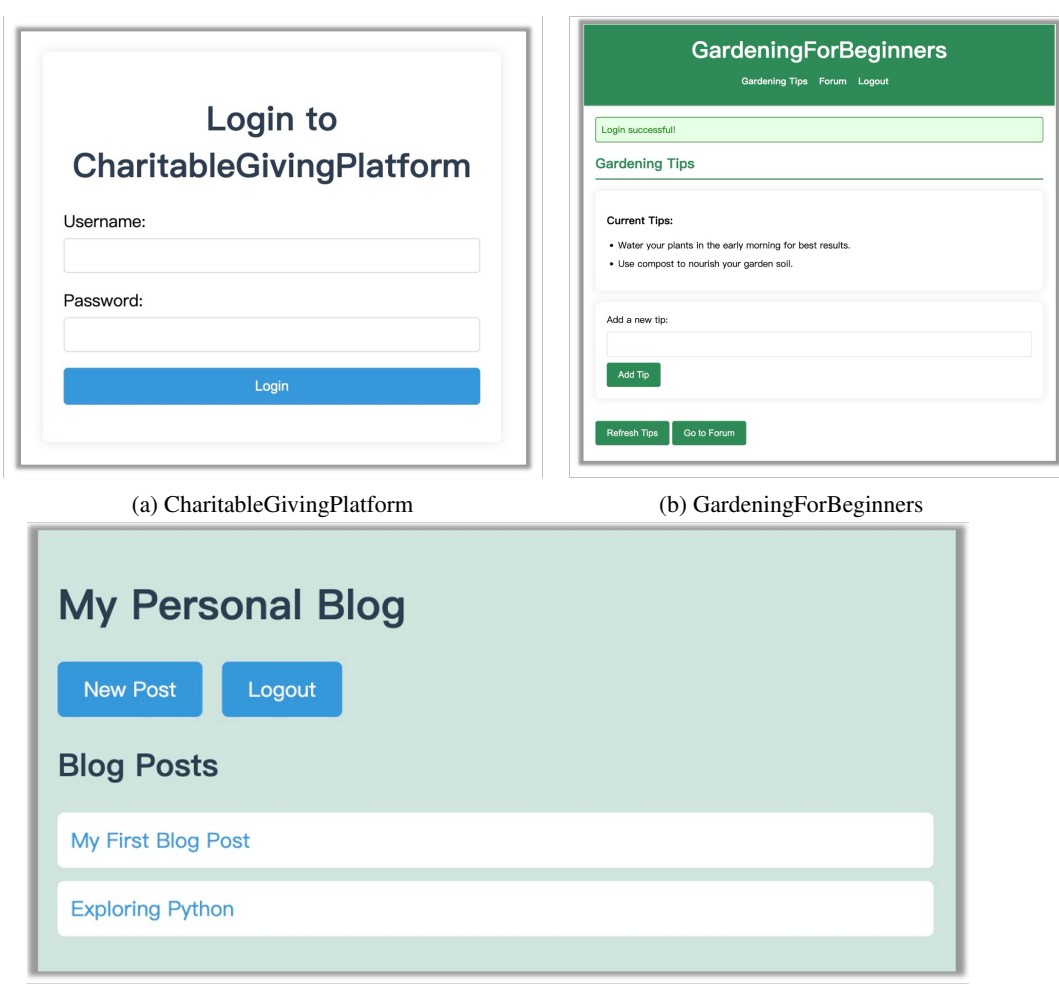

(a) CharitableGivingPlatform        (b) GardeningForBeginners

(c) PersonalBlog

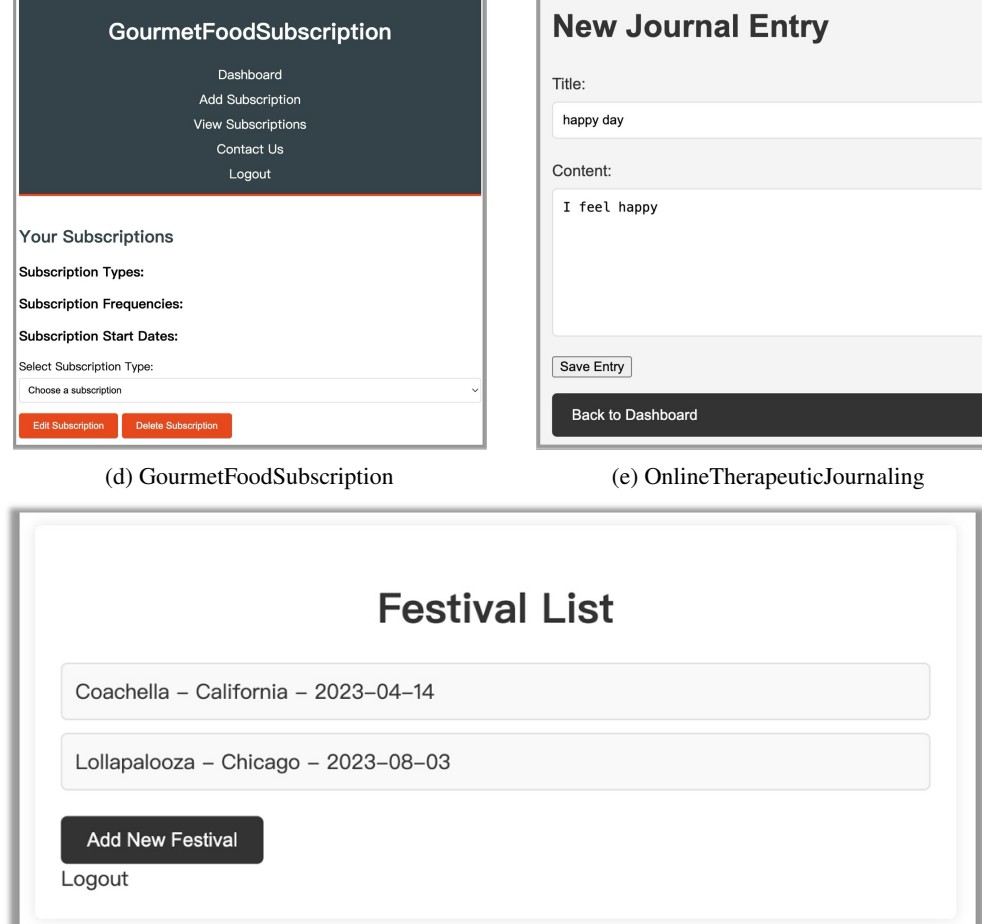

(d) GourmetFoodSubscription          (e) OnlineTherapeuticJournaling

(f) MusicFestivalDirectory

Figure 13: Websites generated by EvoMAC.

