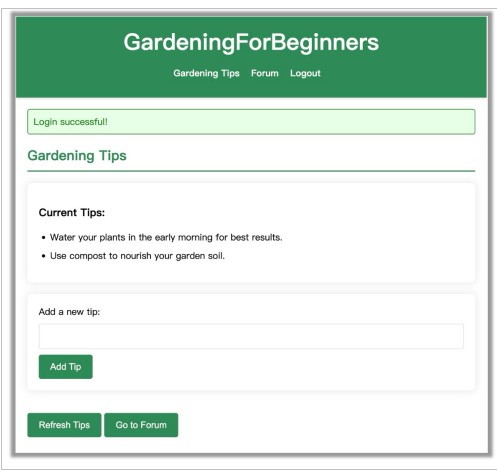

(a) CharitableGivingPlatform                  (b) GardeningForBeginners

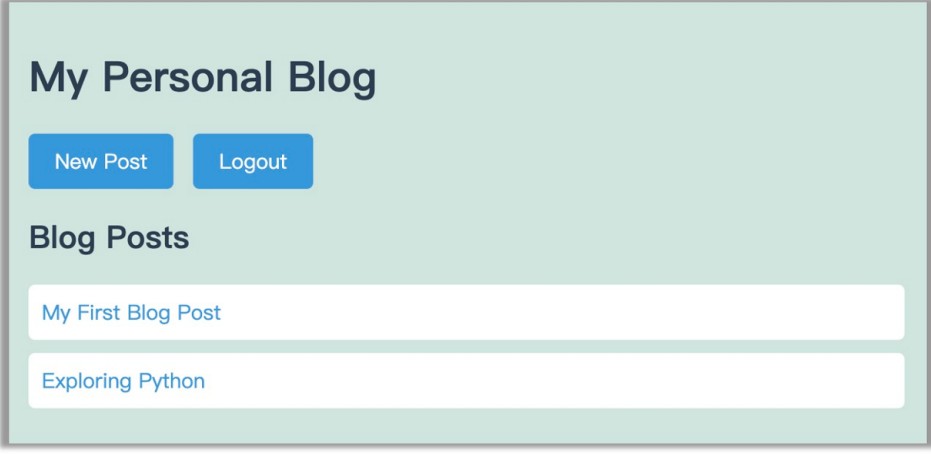

(c) PersonalBlog

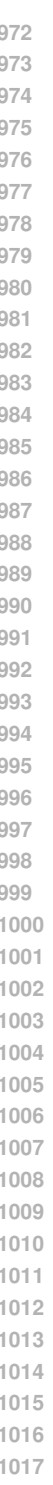

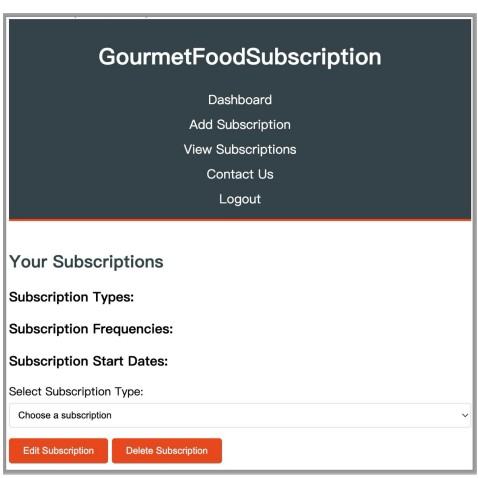

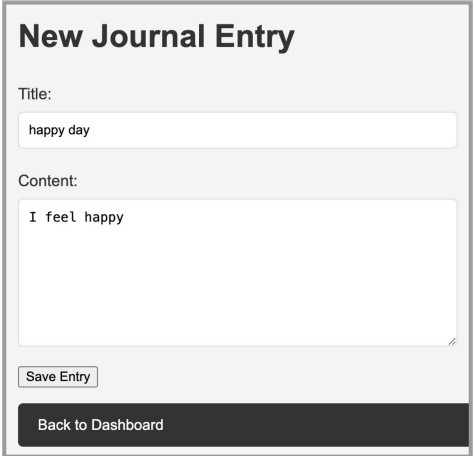

(d) GourmetFoodSubscription

(e) OnlineTherapeuticJournaling

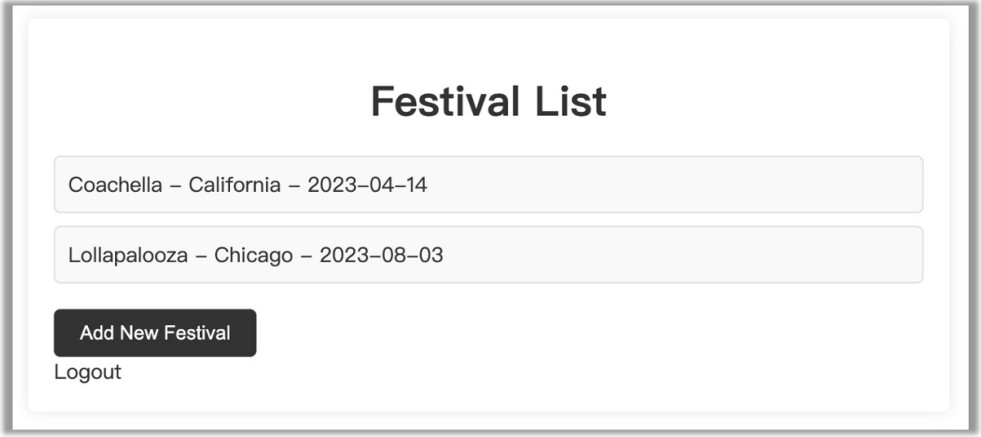

(f) MusicFestivalDirectory

Figure 5: Websites generated by EvoMAC.

# REFERENCES

Mark Chen, Jerry Tworek, Heewoo Jun, Qiming Yuan, Henrique Ponde de Oliveira Pinto, Jared Kaplan, Harri Edwards, Yuri Burda, Nicholas Joseph, Greg Brockman, Alex Ray, Raul Puri, Gretchen Krueger, Michael Petrov, Heidy Khlaaf, Girish Sastry, Pamela Mishkin, Brooke Chan, Scott Gray, Nick Ryder, Mikhail Pavlov, Alethea Power, Lukasz Kaiser, Mohammad Bavarian, Clemens Winter, Philippe Tillet, Felipe Petroski Such, Dave Cummings, Matthias Plappert, Fotios Chantzis, Elizabeth Barnes, Ariel Herbert-Voss, William Hebgen Guss, Alex Nichol, Alex Paino, Nikolas Tezak, Jie Tang, Igor Babuschkin, Suchir Balaji, Shantanu Jain, William Saunders, Christopher Hesse, Andrew N. Carr, Jan Leike, Josh Achiam, Vedant Misra, Evan Morikawa, Alec Radford, Matthew Knight, Miles Brundage, Mira Murati, Katie Mayer, Peter Welinder, Bob McGrew, Dario Amodei, Sam McCandlish, Ilya Sutskever, and Wojciech Zaremba. Evaluating large language models trained on code. 2021.