# OpenReview forum: "Self-Evolving Multi-Agent Collaboration Networks for Software Development"
_ICLR.cc/2025/Conference — ICLR 2025 Poster_

### Official Review · Reviewer_sExs · 2024-10-30

**Soundness:** 3
**Presentation:** 4
**Contribution:** 3
**Rating:** 8
**Confidence:** 4

**Summary:**

The authors provide a paper with twofold contribution: i) a new approach to developing multi-agent collaboration systems for software development and ii) a benchmark to compare their approach with existing approaches to solving the same problem.
Regarding (i), their approach (called EvoMAC) intends to overcome the limitations of similar multi-agent collaboration systems development workflows, mainly on adaptability and generalization. EvoMAC was designed to mimic standard neural network development, meaning that the errors are "backpropagated" throughout the agents, creating a self-adaptive multi-agent network.
Regarding (ii), the benchmark dataset RSD-bench was created based on the requirements of the software being developed, in contrast to the existing ones, which are usually based on the functionality of the generated code/software (i.e., unit tests).
The paper's results show that the EvoMAC approach outperforms other approaches when applying the RSD-Bench to adapted versions of standard benchmark datasets like HumanEval.

**Strengths:**

The paper is very well written and clearly explained. The two proposed contributions (the EVoMAC approach and RSD-Bench dataset) are consistent with the objective expressed in the introduction.
Altough the problem of how to (self-)organize multi-agents is not exactly a new problem, the rise of agentic approaches using LLMs brought new traction to this challenge, and the authors addressed a pain point on these approaches when dealing with complex software development. Indeed, most solutions rely only on function-level and ignore the requirements engineering perspective, which ultimately leads the developers to half-baked solutions. This is not the case with the proposed EvoMAC. It takes account of the particularities of user requirements when organizing the agents initially and also considers the evaluation of the generated code against initial requirements.
Their inspiration for neural network algorithms is broad. However, it is also a clever idea that sounds original and demonstrates a creative adaptation of backpropagation principles to multi-agent collaboration.
The authors also provide sound experimentation on how they implement their approach and when comparing with similar solutions.
Together with the approach, the authors provided a well-defined benchmark to overcome the limitations of the existing ones. A more detailed description of the RSD-bench could indeed compose a contribution per se.
The RSD bench is tailored to requirements engineering, which could address common limitations in agent-based collaboration research by bridging functionality-based assessments with requirement-based evaluations.

**Weaknesses:**

Most of the paper's weaknesses are minor problems that can improve its quality, even though I don't consider them mandatory.
- I truly believe that the mathematical explanation of the problem (mainly between lines 169-184) is unnecessary, even though I see that this somehow facilitates some later explanations. Altough it provides some kind of generalization, the approach still relies on LLMs that are probabilistic by nature and then are not mathematically generalizable (i.e., the function $\phi(\cdot,\cdot)$ is essentially a message sent to an LLM, so it does not behave exactly as a mathematical generic function as the authors may want to demonstrate). I advise the removal of the mathematical explanation. It can help sometimes but does not add value to the paper.
- The paper lacks an actual example/use case in the opposite (or complementary, if the authors decide to keep it) of the mathematical explanation. The authors use this type of explanation in lines 270-271. This kind of example can be used as a running example throughout the paper.
- Minor problems:
	- Typo on the Figure 3 caption. "indrection"
	- Figures 6 and 8 can be improved to ensure legibility and accessibility, maybe by adjusting font size and contrast.
	- Figure 1 is a bit confusing. I think it can be split into 3 different figures or be better explained in the paper. If kept as is, I suggest adding brief explanations of the arrows, such as "add," "revise," and "remove."
	- Figure 5 can be rethought with a better color choice, especially considering the accessibility of the paper.

**Questions:**

- When explaining their approach (section 3.1), the authors did not mention problems regarding the context window of most LLMs. Since requirements can contain quite a large amount of textual data, is the approach capable of dealing with it without extra techniques (e.g., RAG)? If not, even though the authors did not highlight it as a problem, the context window limitations should be mentioned.

---

> ### Author Response · Authors · 2024-11-20
> **Response to Reviewer sExs**
>
> We sincerely appreciate your thoughtful comments. Below, we respond to each point in detail. If our responses adequately address your concerns, we would be truly grateful if you would consider raising your score. We also remain open to any further discussions that could contribute to enhancing the quality of our paper.
>
> [**Weakness 1**]
> I truly believe that the mathematical explanation of the problem (mainly between lines 169-184) is unnecessary, even though I see that this somehow facilitates some later explanations. Although it provides some kind of generalization, the approach still relies on LLMs that are probabilistic by nature and then are not mathematically generalizable (i.e., the function is essentially a message sent to an LLM, so it does not behave exactly as a mathematical generic function as the authors may want to demonstrate). I advise the removal of the mathematical explanation. It can help sometimes but does not add value to the paper.
>
> **Answer:** Sorry for the confusion. The mathematical formulation is incorporated as an objective, drawing an analogy to neural network optimization. This addition aims to help readers better understand the target proxy and its role in motivating the subsequent textual backpropagation-based self-evolving solution. To ensure greater precision and clarity, we will revise this mathematical section to highlight this analogy feature and avoid claiming a mathematical generic function.
>
> ---
>
> [**Weakness 2**]
> The paper lacks an actual example/use case in the opposite (or complementary, if the authors decide to keep it) of the mathematical explanation. The authors use this type of explanation in lines 270-271. This kind of example can be used as a running example throughout the paper.
>
> **Answer:**
> Thanks for the suggestions. We provide a detailed example to clarify each symbol and operation in Tab.4 in the updated appendix.
>
> ---
>
> [**Weakness 3**]
> Typo and visualization suggestions.
>
> **Answer:** Thanks for the suggestions. We will fix the typos and improve the visualization!
>
> ---
>
> [**Question 1**]
> When explaining their approach (section 3.1), the authors did not mention problems regarding the context window of most LLMs. Since requirements can contain quite a large amount of textual data, is the approach capable of dealing with it without extra techniques (e.g., RAG)? If not, even though the authors did not highlight it as a problem, the context window limitations should be mentioned.
>
> **Answer:**
> EvoMac can manage extensive requirements without relying on additional techniques. Unlike a single agent, EvoMAC is much less affected by the context window limitations.
>
> 1. EvoMAC achieves this through the multi-agent collaboration approach. Multi-agent collaboration effectively mitigates context window limitation issues by breaking down complex and lengthy requirements into smaller, more manageable subtasks. These subtasks fit within the context window of individual agents, enabling them to address specific aspects of the task. Gradually, the collective efforts of multiple agents allow EvoMac to fulfill the overall extensive task requirements.
> 2. Figure 7 shows that a single agent fails when requirements become too lengthy, due to the limited context window. In contrast, EvoMAC (powered by the same language model) maintains superior performance even as requirements lengthen. This demonstrates that the context window limitations do not severely impact EvoMAC, allowing it to effectively address longer requirements.

---

> > ### Comment · Reviewer_sExs · 2024-11-27
> >
> > Thank you for your responses. My questions have been addressed, and I will keep my score and acceptance. I repeat my comments on the visualization problems for accessibility, but they don't affect my original score.

---

> ### Comment · Area_Chair_ZWhA · 2024-11-24
> **From AC.**
>
> Reviewer sExs: if possible, can you reply to the rebuttal?

---

### Official Review · Reviewer_aNRL · 2024-11-03

**Soundness:** 4
**Presentation:** 4
**Contribution:** 3
**Rating:** 8
**Confidence:** 3

**Summary:**

The paper presents EvoMAC, a self-evolving multi-agent collaboration (MAC) network designed to advance LLM-based multi-agent systems beyond function-level tasks to software-level coding tasks. EvoMAC employs a unique textual backpropagation mechanism to iteratively update agents and their connections in response to text-based environmental feedback, effectively enhancing task completion without human intervention. By formulating the evolving process to analogize neural network training, EvoMAC provides a clear structure for defining and extracting improvements. This approach underscores the significance of iterative refinement in the software generation process, enabling continuous improvement and adaptability in complex coding tasks.

To evaluate EvoMAC, the authors introduce RSD-Bench, a novel benchmark with complex and diverse software requirements that includes automated requirement verification. EvoMAC demonstrates superior performance on RSD-Bench and the HumanEval function-level benchmark, outperforming state-of-the-art methods and showcasing its robustness and adaptability across various evolving iterations and LLM configurations.

**Strengths:**

- Effectively demonstrates EvoMAC's evolution process as analogous to neural network training, establishing a reliable target proxy for evaluation and constructing a clear objective.

-  The paper is well-structured and easy to follow, with thorough explanations of EvoMAC’s self-evolving process,the design of the RSD-Bench, and detailed descriptions of experimental procedures. Figures and benchmarks illustrate the methodology effectively, aiding comprehension.

- By addressing limitations in traditional MAC systems and demonstrating EvoMAC’s efficacy on challenging software-level tasks, this work sets a promising precedent for adaptive agent frameworks in automated software development, making it valuable for both research and practical applications.

- The RSD-Bench is more practical in software generation evalaution, as it aligns closely with real-world software development process. By incorporating unit tests at both the task and function levels, it establishes a rigorous and precise mechanism for evaluating software generation quality. Additionally, RSD-Bench demonstrates strong human alignment (0.9922), providing a reliable evaluation metric. This paper also conducts analysis the reasonality of RSD-Bench in comparison to existing benchmarks, demonstrates its necessity.

- This paper provides thorough experiments and analyses that robustly demonstrate EvoMAC’s effectiveness and performance. EvoMAC’s strong performance on both the RSD-Bench and HumanEval benchmarks highlights its high quality and efficacy in handling complex coding tasks.

**Weaknesses:**

- Existing studies, such as [1-4] have also explored automation in LLM-based multi-agent collaboration. Please compare the differences between EvoMAC and these works.

- EvoMAC's updating process includes removing agents that have completed their tasks. Can the entire agentic workflow be replayed once a task is finished, or is the removed agent permanently excluded from further iterations?

- Given that EvoMAC includes multiple evolutionary iterations, direct comparisons with standard multi-agent frameworks may not be entirely fair. Could you also provide the number of LLM calls for tasks in RSD-Bench? This metric would offer a more clearer understanding of EvoMAC’s performance.

-  EvoMAC primarily focuses on models like GPT-4, Claude 3.5, and Gemini, but it is unclear if the framework can adapt to less powerful models, such as GPT-3.5 or open-source options like DeepSeek. Presenting results across a broader range of LLMs would support EvoMAC’s claims of robustness and adaptability.

- Can the authors provide additional examples of the unit tests designed within RSD-Bench?

- The Testing Team’s performance significantly impacts the Coding Team's potential, particularly in the HumanEval benchmark. How is the Testing Team’s performance evaluated to ensure alignment with target performance objectives and to prevent divergence? Additionally, how is the Testing Team’s performance quantified within RSD-Bench?

-  The paper does not specify the stopping criteria for EvoMAC’s iterative evolution process. Could the authors provide details on the stopping mechanism or criteria?

- The paper lacks specific settings for the Coding Team in both the HumanEval and RSD-Bench benchmarks. Please provide these details to improve clarity on the experimental configuration and consistency across benchmarks.

- Could the authors showcase additional examples of the textual gradient analysis and the updating process during the evolution for HumanEval and RSD-Bench?



[1] Liu Z, Zhang Y, Li P, et al. Dynamic llm-agent network: An llm-agent collaboration framework with agent team optimization[J]. arXiv preprint arXiv:2310.02170, 2023.

[2] Zhuge M, Wang W, Kirsch L, et al. GPTSwarm: Language Agents as Optimizable Graphs[C]//Forty-first International Conference on Machine Learning.

[3] Hu S, Lu C, Clune J. Automated design of agentic systems[J]. arXiv preprint arXiv:2408.08435, 2024.

[4] Qian C, Xie Z, Wang Y, et al. Scaling Large-Language-Model-based Multi-Agent Collaboration[J]. arXiv preprint arXiv:2406.07155, 2024.

**Questions:**

Please refer to the questions in Weaknesses.

---

> ### Author Response · Authors · 2024-11-20
> **Response to Reviewer aNRL**
>
> We sincerely appreciate your thoughtful comments. Below, we respond to each point in detail. If our responses adequately address your concerns, we would be truly grateful if you would consider raising your score. We also remain open to any further discussions that could contribute to enhancing the quality of our paper.
>
> [**Weakness 1**]
> Existing studies, such as [1-4] have also explored automation in LLM-based multi-agent collaboration. Please compare the differences between EvoMAC and these works.
>
> **Answer:**
> Thanks for the valuable suggestions. Following previous works[1,2,3,4], we model multi-agent collaboration as a network/graph and provide a detailed comparison in the table below. Compared to previous approaches, EvoMAC offers three distinct advantages.
>
> 1. EvoMAC jointly optimizes both nodes and edges, while previous approaches either rely on predefined, non-optimizable solutions [1,3,4] or support only separate optimization of nodes or edges[2]. Predefined collaboration structures are limited by human design, as a single structure cannot adapt to all scenarios. Separate optimization of nodes or edges is also suboptimal because both agent roles (nodes) and their connections (edges) are crucial to task completion. For example, in a coding task, each agent's role and the dependencies among agents are essential; if agents (nodes) are not properly optimized, subtasks may be incorrectly implemented, and if connections (edges) are not optimized, agents may conflict by modifying the same code, resulting in reduced performance. Our joint node and edge optimization offers greater flexibility, enabling more effective multi-agent collaboration.
>
> 2. EvoMAC uses external tools to provide informative, objective textual feedback. In contrast, existing methods [1, 2, 3] primarily rely on scalar feedback or offer no feedback at all [4], as textual feedback is often unavailable for most tasks. Textual feedback, however, not only validates the effectiveness of the system like scalar feedback but also identifies specific errors within the collaboration system and offers actionable guidance to improve multi-agent collaboration.
>
> 3. EvoMAC proposes a novel textual backpropagation method for system optimization while existing approaches rely primarily on heuristic design or scalar-based reinforcement learning (RL) techniques. Heuristic design lacks flexibility, leading to suboptimal performance. RL techniques rely on a single numerical value to optimize the entire complex multi-agent system, making optimization extremely challenging and demonstrating inferior optimization quality. Our novel textual backpropagation method leverages detailed textual feedback and strategically categorizes errors and modification suggestions for each node and edge, making optimization more attainable and demonstrating superior performance.
>
>
> | Method      | Node (Agent)         | Edge (Agent connection) | Feedback                              | Tool         | Optimizer               | Test time evolving |
> | ----------- | -------------------- | ----------------------- | ------------------------------------- | ------------ | ----------------------- | ------------------ |
> | DyLAN[1]    | Predefined           | Optimizable             | Scalar (Heuristical design)           | -            | Heuristical design      | $\checkmark$       |
> | GPTSwarm[2] | Seperately optimized | Seperately optimized    | Scalar (Objective performance)        | -            | RL                      | -                  |
> | ADAS[3]     | Searched             | Predefined              | Scalar (Objective performance)        | -            | Heuristical design      | $\checkmark$       |
> | MacNet[4]   | Predefined           | Predefined              | -                                     | -            | -                       | -                  |
> | EvoMAC      | Jointly optimized    | Jointly optimized       | Text (Objective environment feedback) | $\checkmark$ | Textual backpropogation | $\checkmark$       |
>
> ---
>
> [**Weakness 2**]
> EvoMAC's updating process includes removing agents that have completed their tasks. Can the entire agentic workflow be replayed once a task is finished, or is the removed agent permanently excluded from further iterations?
>
> **Answer:**
> Sorry for the confusion. We will provide more details in the revision. Specifically, the removed agent remains part of the agentic workflow. However, if its subtask is marked as completed, the agent will not be executed in that iteration. This allows the final agentic workflow to be replayed by executing all agents in the workflow as needed.
>
> ---

---

> > ### Author Response · Authors · 2024-11-20
> > **Response to Reviewer aNRL**
> >
> > [**Weakness 3**]
> > Given that EvoMAC includes multiple evolutionary iterations, direct comparisons with standard multi-agent frameworks may not be entirely fair. Could you also provide the number of LLM calls for tasks in RSD-Bench? This metric would offer a more clearer understanding of EvoMAC’s performance.
> >
> > **Answer:**
> > We compare the average API calls between EvoMAC, ChatDev, and ChatDev* in the table below. ChatDev is the best multi-agent baseline on RSD-Bench in Table 1. ChatDev* is the varaint of ChatDev with larger api call budget. We see that 1) simply increasing the number of api call times can not bring a performance improvement; 2) although our EvoMAC requires relatively more api call times, it leads to significant improvement; 3) for simpler tasks like HumanEval, EvoMAC can organize the team more efficiently and thus 'gets twice the result with half effort'.
> > | Method   | Web-Basic | Web-Advanced | Web (APICall) | Game-Basic | Game-Advanced | Game (APICall) | HumanEval | HumanEval (APICall) |
> > | -------- | --------- | ------------ | ------------- | ---------- | ------------- | -------------- | --------- | ------------------- |
> > | ChatDev  | 62.67     | 43.45        | 7.09          | 53.63      | 32.26         | 14.00          |    70.73       | 12.55               |
> > | ChatDev* | 55.13     | 31.17        | 61.11         | 45.65      | 35.48         |        36.63        | /         | /                   |
> > | EvoMAC   | 89.38     | 65.05        | 47.87         | 77.54      | 51.60         | 57.13          |     94.51      | 8.01                |
> >
> > ---
> >
> >
> > [**Weakness 4**]
> > EvoMAC primarily focuses on models like GPT-4, Claude 3.5, and Gemini, but it is unclear if the framework can adapt to less powerful models, such as GPT-3.5 or open-source options like DeepSeek. Presenting results across a broader range of LLMs would support EvoMAC’s claims of robustness and adaptability.
> >
> > **Answer:**
> > Thanks for the suggestions. We equipped EvoMAC with smaller models, including Qwen2.5-Coder-7B-Instruct and Qwen2.5-Coder-14B-Instruct, and the performance results are shown in the table below.
> >
> > 1. EvoMAC performs effectively with models of varying sizes, ranging from smaller models like 7B and 14B to GPT-4o-mini. Regardless of the model used, EvoMAC consistently outperforms single-agent systems, highlighting its robustness and adaptability.
> >
> > 2. EvoMAC's effectiveness does not demand very high-capacity LLMs. For instance, the 14B model achieves performance gains of 28.36% and 29.94% on Website Basic and Advanced tasks, respectively, comparable to GPT-4o-mini's gains of 26.48% and 20.26%.
> >
> >
> > | Method | Model                      | Web-Basic      | Web-Advanced   |
> > | ------ | -------------------------- | -------------- | -------------- |
> > | Single | Qwen2.5-Coder-7B-Instruct  | 24.32          | 7.57           |
> > | Single | Qwen2.5-Coder-14B-Instruct | 43.25          | 20.65          |
> > | Single | GPT-4o-Mini                | 62.90          | 44.40          |
> > | EvoMAC | Qwen2.5-Coder-7B-Instruct  | 27.74 (+3.42)  | 7.69 (+0.12)   |
> > | EvoMAC | Qwen2.5-Coder-14B-Instruct | 71.58 (+28.26) | 50.61 (+29.94) |
> > | EvoMAC | GPT-4o-Mini                | 89.38 (+26.48) | 65.05 (+20.65) |
> >
> > ---
> >
> > [**Weakness 5**] Can the authors provide additional examples of the unit tests designed within RSD-Bench?
> >
> > **Answer:**
> > Yes! We show the test cases used for evaluation and EvoMAC's generated test cases at Tab 11 and 14 in the appendix, including task prompt(Requirement), subtask decomposed by Test Organizer, evaluation metric test case, and generated test case.

---

> > > ### Author Response · Authors · 2024-11-20
> > > **Response to Reviewer aNRL**
> > >
> > > [**Weakness 6**]
> > > The Testing Team’s performance significantly impacts the Coding Team's potential, particularly in the HumanEval benchmark. How is the Testing Team’s performance evaluated to ensure alignment with target performance objectives and to prevent divergence? Additionally, how is the Testing Team’s performance quantified within RSD-Bench?
> > >
> > > **Answer:**
> > > To address the reviewer's confusion, we provide clarification from the following three perspectives.
> > >
> > > 1. On HumanEval, we manually design prompts for the testing agent to mimic the given test case examples, prioritizing precision to ensure accuracy over completeness. This approach ensures that the test-case-based environment feedback remains highly accurate while minimizing potential negative impacts on code generation. The testing performance is primarily evaluated automatically, with a small portion manually assessed. Given HumanEval's high accuracy, we treat the generated code that passes the evaluation test cases as ground-truth code. Generated test cases are marked as incorrect if the ground-truth code fails to pass them. For cases where the code does not pass, we manually verify the accuracy of the test cases.
> > >
> > > 2. To quantify testing performance within RSD-Bench, we manually assessed the accuracy of the generated test cases to determine if they correctly reflect the requirements. The overall accuracy ranges from approximately 70% to 80%. These test cases are effective in verifying the completeness of requirements. While not perfect, they significantly contribute to the evolution process, enabling notable performance gains of 23.28%, 27.93%, 9.44%, and 9.67% on Web-Basic, Web-Advanced, Game-Basic, and Game-Advanced tasks, respectively.
> > >
> > > 3. To facilitate automatic verification of testing performance, we are developing a more comprehensive benchmark that incorporates ground-truth code. The inclusion of ground-truth code will enable automatic assessment of test case accuracy, as properly implemented code should pass all valid test cases unless the test cases themselves are flawed.
> > >
> > > ---
> > >
> > > [**Weakness 7**]
> > > The paper does not specify the stopping criteria for EvoMAC’s iterative evolution process. Could the authors provide details on the stopping mechanism or criteria?
> > >
> > > **Answer:**
> > > Sorry for the confusion. We will provide more details in the revision. Specifically, the process employs two stopping criteria:
> > > 1. The iteration limit is reached, which is set between 3 and 5 iterations to balance effectiveness and efficiency.
> > > 2. All test cases are successfully passed, signifying that all requirements have been met and no further iterations are needed.

---

> > > > ### Author Response · Authors · 2024-11-20
> > > > **Response to Reviewer aNRL**
> > > >
> > > > [**Weakness 8**]
> > > > The paper lacks specific settings for the Coding Team in both the HumanEval and RSD-Bench benchmarks. Please provide these details to improve clarity on the experimental configuration and consistency across benchmarks.
> > > >
> > > > **Answer:**
> > > > Sorry for the confusions. We will provide more details in the revision.
> > > > 1. The coding team is automatically organized by the Coding Organizer agent (Figure 2). For both the HumanEval and RSD-Bench benchmarks, most of the agent prompts remain consistent, ensuring highly comparable experimental settings. The prompts for the Coding Organizer are provided below.
> > > >
> > > > 2. There are two key differences in the configuration of **default tasks** and **num_agent**. For the default tasks, RSD-Bench requires fundamental capabilities tailored to different software types, such as GUI and logging for games, which are not needed for HumanEval. Regarding **num_agent**, HumanEval focuses on simpler, function-level tasks, so the maximum number of agents is set to 2. In contrast, the more complex RSD-Bench tasks allow for up to 5 agents to accommodate the increased complexity.
> > > >
> > > > > **Coding organzier prompt**
> > > > > According to the new user's task and our software designs listed below:
> > > > > Task: "\{task\}".
> > > > > Task description: "\{description\}".
> > > > > Modality: "\{modality\}".
> > > > > Programming Language: "\{language\}"
> > > > > Requirements analysis: "\{requirements\}"
> > > > > Ideas:"\{ideas\}"
> > > > > Coding plan: "\{codes\}"
> > > > > Your goal is to organize a coding team to complete the software development task.
> > > > > There are two **default tasks: ###**
> > > > > Besides these tasks, you should pay attention to the unachieved requirements and think step by step to formulate the requirements into concrete tasks.
> > > > > You should follow the following format: \"COMPOSITION\" is the composition of tasks, and \"Workflow\" is the workflow of the programmers. Each task is assigned to a programmer, and the workflow shows the dependencies between tasks.
> > > > > \#\#\# COMPOSITION
> > > > > ```
> > > > > Task 1: Task 1 description
> > > > > Task 2: Task 2 description
> > > > > ...
> > > > > ```
> > > > > \#\#\# WORKFLOW
> > > > > ```
> > > > > Task 1: []
> > > > > Task 2: [Task 1]
> > > > > ...
> > > > > ```
> > > > > Please note that the decomposition should be both effective and efficient.
> > > > > 1) Each decomposed task should include the related the functions. The task description should be clear and concise.
> > > > > 2) The composition should be kept as small as possible! (LESS THAN **"\{num\_agents\}"**). If there are more than 5 tasks, consider merging the tasks and focus on the most essential features.
> > > > > 3) The decomposed tasks should fully cover the task definitions.
> > > > > 4) The workflow should not contain circles!
> > > >
> > > > ---
> > > >
> > > > [**Weakness 9**]
> > > > Could the authors showcase additional examples of the textual gradient analysis and the updating process during the evolution for HumanEval and RSD-Bench?
> > > >
> > > > **Answer:**
> > > >
> > > > Sorry for the confusion. We show the updating process on RSD-Bench and HumanEval in the Tab 17 and 20 in the updated appendix. We can see that the updating agent will adjust the job of each coder dynamically according to the result of test team.

---

> > > > > ### Comment · Reviewer_aNRL · 2024-11-26
> > > > >
> > > > > Thank you for the authors' detailed responses, most of my concerns have been addressed, and I will keep my score.

---

> ### Comment · Area_Chair_ZWhA · 2024-11-24
> **From AC.**
>
> Reviewer aNRL: if possible, can you reply to the rebuttal?

---

### Official Review · Reviewer_32VP · 2024-11-04

**Soundness:** 3
**Presentation:** 3
**Contribution:** 2
**Rating:** 6
**Confidence:** 3

**Summary:**

The paper introduces a novel framework called EvoMAC, aimed at enhancing the capabilities of LLM-driven multi-agent collaboration (MAC) systems in software development. The authors argue that traditional MAC systems are heavily reliant on human-designed workflows, which limits their adaptability and performance in real-world scenarios. EvoMAC seeks to overcome these limitations by enabling self-evolution of agents and their connections during task execution.

**Strengths:**

- This self-evolving paradigm allows MAC networks to adapt iteratively based on environmental feedback. The framework employs a mechanism similar to neural network backpropagation, where the output of the MAC network is verified against a target proxy, facilitating continuous learning and improvement.
- The RSD-Bench provides a structured benchmark for software-level coding tasks, focusing on comprehensive requirements rather than isolated functions.
- By incorporating unit tests and compilers as feedback mechanisms, EvoMAC reduces subjectivity and provides reliable feedback, which is critical for verifying the correctness of generated code. The objective environment-based feedback is an effective alternative to critique agents, which can introduce bias and hallucinations.

**Weaknesses:**

- The EvoMAC framework introduces significant complexity with its multi-agent setup, dynamic adjustments, and textual backpropagation mechanism. This complexity may limit the framework's accessibility and implementation ease for real-world adoption outside of specialized research contexts.
- Although EvoMAC performs well with large models like GPT-4o-Mini, its performance with smaller or less capable models is unclear. This reliance may restrict its applicability, particularly in environments with limited computational resources.
- RSD-Bench focuses on website and game software types, which may not comprehensively represent the diversity of real-world software development tasks. Expanding the evaluation to include other domains, such as enterprise applications or data processing software, would enhance the generalizability of the results.

**Questions:**

1. How sensitive is EvoMAC to the quality and specificity of feedback from unit tests? If the unit tests are incomplete or overly general, would EvoMAC still produce reliable code, or would it require stricter validation criteria?
2. Can EvoMAC work effectively with models of different sizes, or does it rely on the power of high-capacity LLMs? Would it perform satisfactorily with smaller models that might be more efficient in constrained environments?
3. Can the self-evolving mechanism be applied to other domains outside software development? If yes, how?
4. Given EvoMAC’s iterative approach, how would it handle larger software projects with thousands of lines of code and extensive requirements? Are there specific design considerations for scaling it to more extensive projects?

---

> ### Author Response · Authors · 2024-11-20
> **Response to Reviewer 32VP**
>
> We sincerely appreciate your thoughtful comments. Below, we respond to each point in detail. If our responses adequately address your concerns, we would be truly grateful if you would consider raising your score. We also remain open to any further discussions that could contribute to enhancing the quality of our paper.
>
> [**Weakness 1**]
> The EvoMAC framework introduces significant complexity with its multi-agent setup, dynamic adjustments, and textual backpropagation mechanism. This complexity may limit the framework's accessibility and implementation ease for real-world adoption outside of specialized research contexts.
>
> **Answer:**
> Thanks for the time in reviewing. To address the reviewer's concerns about EvoMAC's applicability in real-world scenarios, we offer clarification from the following four perspectives.
> 1. In real-world scenarios, software development may require hundreds of engineers working collaboratively over months or even years to achieve completion. The primary focus within the software development community is on improving the effectiveness of the development process, advancing from function-level requirements to more extensive software-level requirements.
> 2. Multi-agent collaboration is a common practice in automatic software development [1,2], often involving tens or even hundreds of agents to enhance performance. EvoMAC's demonstrated effectiveness makes it well-suited to handling the complexities of real-world software development.
> 3. EvoMAC's approach, which includes a multi-agent setup, dynamic adjustments, and a textual backpropagation mechanism, is fully automated. It eliminates the need for human intervention or specific heuristic designs, ensuring it is highly adaptable to real-world scenarios.
> 4. EvoMAC performs effectively even when powered by smaller 14B models, as shown in the table in **Weakness 2 and Question 2**. This makes it feasible for deployment in resource-constrained real-world environments, offering an efficient solution without compromising performance.
>
> [1] Zhuge M, Wang W, Kirsch L, et al. GPTSwarm: Language Agents as Optimizable Graphs[C]//Forty-first International Conference on Machine Learning.
> [2] Qian C, Xie Z, Wang Y, et al. Scaling Large-Language-Model-based Multi-Agent Collaboration[J]. arXiv preprint arXiv:2406.07155, 2024.
>
> ---
>
> [**Weakness 2 and Question 2**]
> Although EvoMAC performs well with large models like GPT-4o-Mini, its performance with smaller or less capable models is unclear. This reliance may restrict its applicability, particularly in environments with limited computational resources. Can EvoMAC work effectively with models of different sizes, or does it rely on the power of high-capacity LLMs? Would it perform satisfactorily with smaller models that might be more efficient in constrained environments?
>
> **Answer:**
> Thank you for your time and valuable suggestions. To address the reviewer's concerns regarding its applicability in constrained scenarios, we equip EvoMAC with smaller models, including Qwen2.5-Coder-7B-Instruct and Qwen2.5-Coder-14B-Instruct, and the performance results are shown in the table below.
>
> 1. EvoMAC performs effectively with models of varying sizes, ranging from smaller models like 7B and 14B to GPT-4o-mini. Regardless of the model used, EvoMAC consistently outperforms single-agent systems, highlighting its generalizability and effectiveness.
>
> 2. EvoMAC's effectiveness does not demand very high-capacity LLMs. For instance, the 14B model achieves performance gains of 28.36% and 29.94% on Website Basic and Advanced tasks, respectively, comparable to GPT-4o-mini's gains of 26.48% and 20.26%. However, the LLM capacity must not be too limited; the 7B model with performance 24.32% and 7.57% shows more modest improvements.
>
> | Method | Model                      | Web-Basic      | Web-Advanced   |
> | ------ | -------------------------- | -------------- | -------------- |
> | Single | Qwen2.5-Coder-7B-Instruct  | 24.32          | 7.57           |
> | Single | Qwen2.5-Coder-14B-Instruct | 43.25          | 20.65          |
> | Single | GPT-4o-Mini                | 62.90          | 44.40          |
> | EvoMAC | Qwen2.5-Coder-7B-Instruct  | 27.74 (+3.42)  | 7.69 (+0.12)   |
> | EvoMAC | Qwen2.5-Coder-14B-Instruct | 71.58 (+28.26) | 50.61 (+29.94) |
> | EvoMAC | GPT-4o-Mini                | 89.38 (+26.48) | 65.05 (+20.65) |
>
> ---

---

> ### Author Response · Authors · 2024-11-20
> **Response to Reviewer 32VP**
>
> [**Weakness 3**]
> RSD-Bench focuses on website and game software types, which may not comprehensively represent the diversity of real-world software development tasks. Expanding the evaluation to include other domains, such as enterprise applications or data processing software, would enhance the generalizability of the results.
>
> **Answer:**
> To address the reviewer's concerns about RSD-Bench's diversity and EvoMAC's generalizability, we offer clarification from the following three perspectives.
>
> 1. Games and websites represent two typical software types, and the testing covers 11 distinct categories. These categories expand beyond previous benchmarks focused on function completion and bug fixing to provide a more comprehensive evaluation of coding capabilities. We are extending the benchmark to include additional software types.
>
> 2. EvoMAC is validated across three distinct software development tasks—function completion, website, and game, where it outperforms prior works, demonstrating its effectiveness and generalizability.
>
> 3. We extend the evaluation of EvoMAC to include data processing tasks by utilizing the InfiAgent-DABench[1] dataset (hard). The table below shows that EvoMAC enhances the performance in handling complex data analysis tasks, showcasing its effectiveness and generalizability to diverse software types.
>
> | Method   | Accuracy |
> | - | - |
> | Single   | 46.50%   |
> | MapCoder | 75.00%   |
> | ChatDev  | 62.50%   |
> | EvoMAC   | 82.50%   |
>
> [1] Hu, Xueyu, et al. "InfiAgent-DABench: Evaluating Agents on Data Analysis Tasks.", ICML 2024
>
> ---
>
> [**Question 1**]
> How sensitive is EvoMAC to the quality and specificity of feedback from unit tests? If the unit tests are incomplete or overly general, would EvoMAC still produce reliable code, or would it require stricter validation criteria?
>
> **Answer:**
>
> To address the reviewer's concerns, we provide clarification from the following three perspectives.
>
> 1. EvoMAC demonstrates relative robustness to unit tests. We randomly sampled 100 generated test cases and manually evaluated their accuracy, as shown in the table below. The results indicate that even when testing accuracy is around 50%, the feedback remains valid and contributes to measurable performance gains.
>
> 2. We manually evaluate testing performance within RSD-Bench, with the overall accuracy ranging from approximately 70% to 80%. These test cases effectively verify the completeness of requirements. Although they do not perfectly cover all code and requirements, they play a significant role in the evolution process, resulting in notable performance gains of 23.28%, 27.93%, 9.44%, and 9.67% on Web-Basic, Web-Advanced, Game-Basic, and Game-Advanced tasks, respectively.
>
> 3. To enable automatic verification of testing performance, we are developing a more comprehensive benchmark that includes ground-truth code. Incorporating ground-truth code will allow for automatic assessment of test case accuracy, as properly implemented code should pass all valid test cases unless the test cases themselves are flawed. This approach will facilitate a more thorough evaluation of testing capabilities and enable adaptive validation criteria, providing more accurate and complete feedback.
>
> |  | Testcase  Accuracy | Performance Gain       |
> | -| - |-|
> | Basic    | 55/72(76.38%)      | +4.35% (69.56%-73.91%) |
> | Advanced | 15/32(46.87%)      | +3.22% (45.16%-48.38%) |
>  ---
>
> [**Question 3**]
> Can the self-evolving mechanism be applied to other domains outside software development? If yes, how?
>
> **Answer:**
> Yes, our self-evolving mechanism can be applied to domains where informative feedback is available. For instance, in the auto-research domain [1,2], which focuses on developing AI agents for automated machine learning tasks, such as dataset creation, algorithm development, and model training. Here, the validation set serves as the target proxy, with feedback provided by validation performance. Analyzing results on the validation set offers valuable insights into the algorithm's effectiveness and can guide further modifications. The self-evolving mechnism iteratively adjusts algorithm generation, collects feedback, and analyzes results, promising advanced automated machine learning performance.
>
> [1] MLE-bench: Evaluating Machine Learning Agents on Machine Learning Engineering, OpenAI 2024
> [2] MLAgentBench: Evaluating Language Agents on Machine Learning Experimentation
>
> ---

---

> > ### Author Response · Authors · 2024-11-20
> > **Response to Reviewer 32VP**
> >
> > [**Question 4**]
> > Given EvoMAC’s iterative approach, how would it handle larger software projects with thousands of lines of code and extensive requirements? Are there specific design considerations for scaling it to more extensive projects?
> >
> > **Answer:**
> > EvoMAC’s multi-agent setup enables automatic scaling for large-scale project development. Experiments in the paper show that EvoMAC effectively expands from function-level code with tens of lines to software-level code with hundreds of lines, with promising potential to further scale to thousands of lines. This scalability is achieved through two key design features:
> >
> > 1. The coding organizer can automatically break down extensive requirements into smaller, manageable sub-tasks and organize a larger team of coding agents to collaboratively develop large-scale software projects. Similarly, it can form a larger testing team to conduct comprehensive testing for these projects.
> >
> > 2. The updating team in EvoMAC continuously adds new agents to handle unfinished requirements, ensuring the completeness of large-scale code generation. Additionally, it removes agents whose tasks are complete, improving development efficiency by avoiding redundant efforts.

---

> ### Comment · Area_Chair_ZWhA · 2024-11-24
> **From AC.**
>
> Reviewer 32VP: if possible, can you reply to the rebuttal?

---

> > ### Comment · Reviewer_32VP · 2024-11-26
> >
> > Thank you for addressing my questions. After reviewing your responses, I have decided to maintain my initial score.

---

> > > ### Author Response · Authors · 2024-11-26
> > > **Response to Reviewer 32VP**
> > >
> > > Thank you for your thoughtful review and for taking the time to consider our responses. We appreciate your engagement and feedback. Could you kindly share the remaining concerns that led to your decision not to raise the score? Understanding this would greatly help us improve our work further. Thank you again for your time and insights.

---

### Official Review · Reviewer_kPYh · 2024-11-04

**Soundness:** 3
**Presentation:** 4
**Contribution:** 3
**Rating:** 6
**Confidence:** 3

**Summary:**

This paper proposes a multi-agent collaboration approach to address software development problems. EvoMAC obtains text-based environmental feedback by verifying the match between the MAC network's output and the target proxy, and it updates the network using a novel textual backpropagation technique, thereby achieving the final development outcome. Additionally, this paper introduces a software development benchmark called RSD-Bench, which provides more detailed and structured software requirements for documenting user needs compared to previous benchmarks. The final experimental results show that the proposed EvoMAC outperforms other single-agent and multi-agent methods on both RSD-Bench and HumanEval.

**Strengths:**

1. This paper proposes a multi-agent collaboration approach to address software development problems.
2. This paper introduces a software development benchmark called RSD-Bench, which provides more detailed and structured software requirements for documenting user needs compared to previous benchmarks.
3. Extensive experiments demonstrate that EvoMAC outperforms other single-agent and multi-agent methods on both RSD-Bench and HumanEval.

**Weaknesses:**

1. The benchmark proposed in this paper lacks data analysis and some basic statistical information, such as prompt length, the number of final generated files/functions, etc.
2. The benchmark proposed in this paper is relatively easy, with the EvoMAC method already achieving around 90% accuracy.

**Questions:**

1. Are there any issues with the citation format in the paper?
2. Does the paper lack an appendix?
3. In Table 2, there is no comparison of results with environment tools but without evolving. This should be added.

---

> ### Author Response · Authors · 2024-11-20
> **Response to Reviewer kPYh**
>
> We sincerely appreciate your thoughtful comments. Below, we respond to each point in detail. If our responses adequately address your concerns, we would be truly grateful if you would consider raising your score. We also remain open to any further discussions that could contribute to enhancing the quality of our paper.
>
> [**Weakness 1**]
> The benchmark proposed in this paper lacks data analysis and some basic statistical information, such as prompt length, the number of final generated files/functions, etc.
>
> **Answer:**
> Thank you for the suggestions. We provide more detailed data analysis and statistical information in the following parts.
> 1. The data analysis of the benchmark is in the supplementary material: Table 1 and Figure 1. Note that the Supplementary Material is submitted as a separate PDF, which can be downloaded by clicking the down-arrow button labeled "Supplementary Material" (below the abstract). For convenience, we also provide a copy of Table 1 below.
>     - Table 1 presents the number of software samples, test cases, and the average/maximum lengths of requirement prompts for both websites and games. We see that: i) our software-level benchmark is diverse, encompassing two common types of software (Websites and Games) as well as two levels of difficulty (Basic and Advanced); and ii) our software-level benchmark is notably challenging, with an average prompt length exceeding 500 on Game and 1000 on Website, making it almost 5 and 10 times longer than the function-level HumanEval benchmark separately.
>
>     - Figure 1 illustrates the distribution of test case types across both categories. We see that our software-level benchmark includes 11 distinct test case categories and a total of 616 test cases, representing a scale 6 times larger than the function-level HumanEval benchmark. This extensive set enables a more focused and thorough evaluation of code generation capabilities.
>
> Table 1:  Basic statistics for website and game domains, including the amount of samples, the task prompt length (Average/Max), and number of test cases at both Basic and Advanced levels.
>
> | Benchmark | Amount | Prompt length(token)| Testcase(Basic) | Testcase(Advanced) |
> |-|-|-|-|-|
> |Website|45| 1011/1553|292|247|
> |Game| 8| 507/788|46 |31|
> |Humaneval|164|131/398|/|/|
>
> 2. We provide more detailed statistical information in the table below, including code length, file count, and function count. We see that the generated code consists of over six functions across multiple files, showcasing advanced capabilities in generating complex, realistic code that requires function coordination and synchronization across files. This suggests that: i) our software-level benchmark closely reflects the advanced coding skills needed for real-world coding tasks, and ii) the proposed multi-agent collaboration system effectively supports the generation of more complex code.
>
>
> |Statistical Information| Game | Website |
> |-|-|-|
> |Generated function count| 6.25$\pm$ 1.30| 10.67$\pm$ 3.39|
> | Generated file count| 1.38$\pm$ 0.70 | 6.53$\pm$ 1.61|
> | Generated code length| 158$\pm$ 17.13 | 276$\pm$ 227.72|
>
> ---
>
> [**Weakness 2**]
> The benchmark proposed in this paper is relatively easy, with the EvoMAC method already achieving around 90% accuracy.
>
> **Answer:**
> To address the reviewer's concern, we would like to clarify that the benchmark is challenging from two aspects.
>
> 1. Our benchmark includes two difficulty levels: Basic and Advanced. While the Basic level achieves around 90% accuracy, the Advanced level proves challenging, with accuracy around 50%.
>
> 2. The Advanced level reflects more complex software functionalities, such as game logic rules and dynamic web content management. Passing each advanced test case is challenging and demands a fundamental improvement in coding capabilities. As meeting a single advanced requirement involves synchronized implementation and function calls across multiple files. For instance, fulfilling the advanced requirement of "eating a mushroom and earning points in Mario" requires checking the locations of the mushroom and Mario, adjusting the game score accordingly, and updating the visualization. It requires implementing multiple interdependent functions, demanding advanced coding capabilities to avoid conflicts.
>
>
> ---
>
> [**Question 1**]
> Are there any issues with the citation format in the paper?
>
> **Answer:**
> Thanks for pointing out. We will fix it.
>
> ---
>
> [**Question 2**]
> Does the paper lack an appendix?
>
> **Answer:**
> We apologize for any confusion regarding the appendix. The appendix has been submitted as a separate PDF and can be accessed by clicking the down-arrow button labeled "Supplementary Material" below the abstract.
>
> ---

---

> > ### Author Response · Authors · 2024-11-20
> > **Response to Reviewer kPYh**
> >
> > [**Question 3**]
> > In Table 2, there is no comparison of results with environment tools but without evolving. This should be added.
> >
> > **Answer:**
> >
> > Thank you for the suggestions. We would like to provide clarification from two aspects.
> >
> > 1. The results with environmental tools but without evolution are shown in Figure 6, representing the performance when the evolving time is set to 0.
> >
> > 2. We have included the results in Table 2. The data demonstrates that evolving leads to performance gains of 23.28%, 27.93%, 9.44%, and 9.67% on Web-Basic, Web-Advanced, Game-Basic, and Game-Advanced, respectively.
> >
> > ||Coding|Testing|Evol.| Env.| Web-Basic| Web-Advanced | Game-Basic | Game-Advanced |
> > |-|-|-|-|-|-|-|-|-|
> > |g|Multi|Multi|$\checkmark$|$\checkmark$|90.75|67.20|77.54|51.60|
> > |h|Multi|Multi|-|$\checkmark$|67.47|39.27|68.10|41.93|

---

> ### Comment · Area_Chair_ZWhA · 2024-11-24
> **From AC.**
>
> Reviewer kPYh: if possible, can you reply to the rebuttal?

---

### Meta-Review · Area_Chair_ZWhA · 2024-12-20

**Metareview:**

The work describes an approach to automated software development. The approach is tested on one benchmark that the paper introduces (RSD-Bench) and on one preexisting benchmark (HumanEval).

The main strength of the submission is that the approach works well in practice, on the two benchmarks.

The paper has the following weaknesses.
- I am not sure the paper is in scope for ICLR. Because of the nature of the paper where it focuses on one application only, I am wondering if perhaps a more applied conference might be a better fit.
- I am not sure if the paper is reproducible. The paper is heavily empirical so this is even more key than normally. There doesn't seem to be code.
- The paper isn't clear enough. For example, it uses the concepts of "textual gradient" and "gradient analysis operator" without first properly defining them in the paper text. From the perspective of an ICLR audience, such lack of rigour is a problem.
- I am not sure why one needs a multi-agent framework to solve a single-agent task (even though I accept the approach works well).

Since reviewers are convinced about the quality of the paper, I recommend acceptance.

**Additional Comments On Reviewer Discussion:**

I think that the paper has fundamental weaknesses and should be rejected despite the positive reviews (see list of weaknesses in the metareview).

I tried discussing this with the reviewers, but didn't get any replies.

I am reluctantly recommending acceptance since I don't think I can overrule all reviewers.

---

### Decision · Program_Chairs · 2025-01-22

Accept (Poster)